



# Significant contrasts in aerosol acidity between China and the Unites States

Bingqing Zhang[1], Huizhong Shen[1], Pengfei Liu[2], Hongyu Guo[3], Yongtao Hu[1], Yilin Chen[1], Shaodong Xie[4], Ziyan Xi[4], Armistead G. Russell[1]

[1]School of Civil and Environmental Engineering, Georgia Institute of Technology, Atlanta, Georgia 30332, USA

[2]School of Earth and Atmospheric Sciences, Georgia Institute of Technology, Atlanta, Georgia 30332, USA

[3]Cooperative Institute for Research in Environmental Sciences and Department of Chemistry, University of Colorado Boulder, Boulder, Colorado 80309, USA

[4]College of Environmental Sciences and Engineering, State Key Joint Laboratory of Environmental Simulation and Pollution Control, Peking University, Beijing, 100871, PR China

*Correspondence to*: Huizhong Shen (hshen73@gatech.edu)

**Abstract.** Aerosol acidity governs several key processes in aerosol physics and chemistry, thus affecting aerosol mass and composition, and ultimately the climate and human health. Previous studies have reported the aerosol pH separately in China and the United States, implying a different aerosol acidity between these two countries. However, underlying mechanisms responsible for the pH difference are not fully understood, limited by the scarcity of simultaneous measurements of aerosol composition and gas species, especially in China. Here we conduct a comprehensive assessment of the aerosol acidity in China and the United States, using extended ground-level measurements and regional chemical transport model simulations. We show aerosol in China is significantly less acidic than that in the United States, with pH values 1–2 units higher. Based on a multivariable Taylor Series method and a series of sensitivity tests, we identify several major factors leading to the pH difference. Compared to the United States, aerosols in China are generally in total ammonia ($TNH_3=NH_4^++NH_3$) rich conditions where particle phase ammonium ($NH_4^+$) concentrations are adequate enough to nearly neutralize major acidic inorganic anions such as sulfate, nitrate, and chloride, leading to a higher aerosol pH. Higher relative availability of the stronger acidic component, sulfate, compared with the weaker acidic component, total nitrate ($TNO_3=NO_3^-+HNO_3$), also contributes to the lower aerosol pH in the United States. As a response to higher aerosol pH, the higher nitrate to sulfate molar ratios in China indicates a nitrate-rich condition, further leading to higher aerosol water uptake which will continually promote nitrate aerosol formation. Considering the historical emissions trends, the difference in aerosol acidity between these two countries is expected to continue as $SO_2$ and $NO_x$ emissions are further controlled. The differences in aerosol acidity highlight in the present study imply potential differences in formation mechanisms, physicochemical properties, and toxicity of aerosol particles between China and the United States.



## 1 Introduction

As an intrinsic aerosol property, aerosol acidity (usually measured by aerosol pH) plays an important role in a variety of aerosol physical and chemical processes (Pye et al., 2020). Aerosol acidity can modulate aerosol mass by controlling the gas-particle partitioning of volatile and semi-volatile acids (such as $HCl-Cl^-$ and $HNO_3-NO_3^-$)(Guo et al., 2016) and influencing production rates of secondary aerosol from heterogeneous pathways (Jang et al., 2002;Surratt et al., 2010;Pathak et al., 2011). It also affects aerosol optical properties via proton dissociation of organic functional groups (Mo et al., 2017) and the morphology or phase state of organic aerosols (Losey et al., 2016;Losey et al., 2018). Recent evidence links aerosol acidity to aerosol toxicity and health outcomes. Highly acidic aerosols, for example, dissolve more metals which can generate reactive oxygen species in vivo (Fang et al., 2017). High aerosol acidity is associated with increased risks of respiratory disease and cancer (Kleinman et al., 1989;Gwynn et al., 2000;Behera et al., 2015).

Due to the difficulties in direct measurements of aerosol pH (Jang et al., 2002;Li and Jang, 2012), thermodynamic models, including ISORROPIA-II (Fountoukis and Nenes, 2007), E-AIM (Clegg et al., 1998), and SCAPE2 (Kim and Seinfeld, 1995), have been widely used to calculate aerosol pH from measured gaseous and aerosol composition and meteorological data such as relative humidity and temperature. A large number of studies suggest that these models are capable of reproducing the partitioning of semi-volatile species including $HNO_3-NO_3^-$ and $NH_4^+-NH_3$, which are sensitive to aerosol pH (Guo et al., 2015;Hennigan et al., 2015;Guo et al., 2016).

Field observations in different regions of the United States indicated that the aerosol acidity was typically high. For example, (Weber et al., 2016) showed aerosol pH in the Southeastern United States was buffered to be nearly constant in the rage of 0-2 despite a substantial sulfate reductions over the past 15 years, and this trend may be applicable to many other regions. Studies in the Northeastern United States and California also indicated a highly acidic aerosol with mean pH values of 0.8 and 1.9, respectively(Guo et al., 2017a). The aerosol pH in the Midwest United States was typically higher than other areas., with the average aerosol pH was 3.8 (Lawal et al., 2018). Studies in China, on the other hand, showed generally higher levels of aerosol pH, indicating a lower aerosol acidity. Several studies in the heavily polluted North China Plain (NCP) region, reported average pH values between 3.5–5.2 (Shi et al., 2017;Ding et al., 2019;Shi et al., 2019;Song et al., 2019;Wang et al., 2020a). Xi'an, a city located in Northwest China, had aerosol pH values up to 5 (Wang et al., 2016;Guo et al., 2017b). Some sites in Southeast China showed a lower aerosol pH, such as the site in Guangzhou with an average value of 2.3 (Jia et al., 2020). A comprehensive, nationwide comparison of aerosol pH between China and the United States can give us a better understanding of the driving factors of aerosol pH and its effect in aerosol formation mechanisms and properties (Pathak et al., 2009;Guo et al., 2017a;Wang et al., 2020a). However, such comparisons are still scarce (Guo et al., 2017b;Nenes et al., 2020), primarily limited by a lack of extensive simultaneous measurements of aerosol composition and semi-volatile gaseous compounds in China.

In this study, we compared the aerosol composition and acidity between China and the United States based on one-year large-scale measurements from 34 ground monitoring sites in the United States and 16 sites in China. In order to extend the spatial coverage to nationwide scales, we employed the Community Multiscale Air Quality (CMAQ) model to simulate the concentrations of gaseous and aerosol species which were used to calculate aerosol pH in both countries. We then carried out a series of sensitivity tests to identify and discuss the causes and effects of the pH difference.

## 2 Data collection and method

### 2.1 Observational data

Gaseous species (including $HNO_3$, $NH_3$ and $HCl$), aerosol components (including $PM_{2.5}$ components of $SO_4^{2-}$, $NO_3^-$, $NH_4^+$, and $Cl^-$), and nonvolatile cations (NVCs) (including $Na^+$, $Mg^{2+}$, $K^+$, and $Ca^{2+}$) collected from monitoring networks in China and the United States and used for analysis and comparison in this study. The names and locations of the monitoring sites can be found in Tables S1 and S2. The sum of total observed aerosol ionic compounds is defined as water soluble ions (WSI), though it is recognized that not all ions are routinely measured, including trace species and organic ions. We also study the partitioning of semi-volatile species including $NH_3$- $NH_4^+$ and $HNO_3$-$NO_3^-$ because they are sensitive to pH, especially when the partitioning ratios, $\varepsilon(NH_4^+)$ and $\varepsilon(NO_3^-)$, defined as the molar ratio of $NH_4^+$ to total ammonia ($TNH_3 = NH_3 + NH_4^+$) and the molar ratio of $NO_3^-$ to total nitrate ($TNO_3 = HNO_3 + NO_3^-$), are around 50%(Guo et al., 2017a;Chen et al., 2019).

In the United States, observational data are from the co-located sites of Clean Air Status and Trends Network (CASTNET) (https://www.epa.gov/castnet) and Ammonia Monitoring Network (AMoN) (http://nadp.slh.wisc.edu/amon/). Two sites in CASTNET and AMoN are assumed to be co-located if within 1km, and then combined for pH calculation. Weekly ambient concentrations of gases and particles including $HNO_3$, $SO_4^{2-}$, $NO_3^-$, $NH_4^+$, $Cl^-$ and NVCs are from CASTNET sites, while biweekly concentrations of $NH_3$ are from AMoN sites. In order to match biweekly data of $NH_3$ from AMoN to weekly data of other species from CASTNET, we assume concentrations of $NH_3$ in two adjacent weeks to be the same. This assumption is expected to have a minor effect on pH prediction as previous study found that 10 times increase in $NH_3$ is required to increase pH by one unit (Guo et al., 2017b) and is also confirmed in the later discussion (Sect.3.2.3). HCl data is not available, so we only use particle phase $Cl^-$ as total Cl in pH calculation. Sensitivity tests either assume a four times of HCl vs. $Cl^-$ concentrations or use derived HCl concentrations from CMAQ modeled $HCl/Cl^-$ ratios show little difference in aerosol pH, compared to that using particle phase $Cl^-$ as total Cl (Fig. S1). Considering the reported little change in aerosol pH in the United States over a long-term period (Lawal et al., 2018;Weber et al., 2016) and the configuration of the chemical transport model which is set up for the year 2011 (see the following section), we use observational data in 2011 to investigate the aerosol pH in the United States. Only the sites with measurements available for all species were selected in this study. As a result, there are 34 co-located sites, which are evenly distributed across the United States (Fig. S2a). The accuracy for CASTNET measurements

were all in 95%-105% (except for $NH_4^+$, which is in 90%-110%), while the accuracy of $NH_3$ data derived from AMoN was in 97%-103%, suggesting a good quality of this datasets. Detailed information about data quality could be found at CASTNET Quality Assurance Report-Annual 2011 (United States Environmental Protection Agency, 2012) and Quality Assurance

Support for the NADP (National Atmospheric Deposition Program).

In China, hourly observational data are extracted from the Data-sharing platform by Comprehensive Observation Network for Air Pollution in Beijing-Tianjin-Hebei and Its surrounding Areas (http://123.127.175.60:8765/siteui/index). This observation network gathers information from multi-source and provides simultaneous observations of gaseous and aerosol species at individual monitoring sites (Wang et al., 2019). In this study, we derive daily average concentrations of gaseous species

including $NH_3$, $HNO_3$ and HCl and of particle species including $NH_4^+$, $NO_3^-$, $Cl^-$ and NVCs for pH calculation from hourly observational data. Due to lack of data quality information, we processed the data before using by removing unreasonable data for quality control. For example, data points showing extremely large values for certain species (e.g. $[Ca^{2+}]>1000\mu g\cdot m^{-3}$) are removed. 16 monitoring sites with measurements available for all species are selected in this study. These sites cluster in NCP in eastern China (Fig. S2c).

**2.2 Model configuration**

We use the CMAQ version 5.0.2 (United States Environmental Protection Agency, 2014) to simulate gaseous and aerosol species concentrations and aerosol pH in China and the United States. The model domains of the two simulations cover the mainland China and the contiguous United States with 124×184 and 112×148 horizontal grid cells, respectively, and are both resolved at the 36-km horizontal resolution and 13 vertical layers extending to ~16 km above the ground. In both simulations,

gas-phase chemistry is modeled with the CB05 chemical mechanism (Yarwood et al., 2005), and the aerosol thermodynamic equilibrium is modeled with ISORROPIA II (Fountoukis and Nenes, 2007).

The meteorological and emission inputs used to drive China's simulation are adopted from "AiMa", an online operational air quality forecasting system (Lyu et al., 2017;AiMa Forecast, 2017). In the AiMa modeling system, the meteorological data were generated with the Weather Research and Forecasting (WRF) model (William C. Skamarock 2008) driven by the 0.5-

degree global weather forecast products produced by the National Centers for Environmental Prediction (NCEP) Global Forecast System (GFS) (Global Forecast System (GFS) Model). The AiMa emission inventory was compiled and derived by integrating a variety of inventories and utilizing various activity data and has been continuously updated since established (Lyu et al., 2017). The base year of the current AiMa emission inventory is 2017. For the simulation in the United States, we use WRF-modeled meteorological fields downscaled from the North American Regional Reanalysis (NARR) data (Mesinger et

al., 2006) as the meteorological input and the 2011 National Emissions Inventory provided by the United States Environmental Protection Agency as the emission input (United States Environmental Protection Agency). The base year of the meteorology



and emissions is consistent with the year of the measurements in each country (i.e., 2017 for China and 2011 for the United States).

In order to evaluate model performance against observations, we calculate normalized mean bias (NMB) and normalized root-mean-square errors (NRMSE) to evaluate the spatial variation of pH, species concentrations and partitioning ratios with following equations,

$$NMB = \frac{\sum_1^N (C_m - C_o)}{\sum_1^N C_o}$$

(1)

$$NRMSE = \frac{\sqrt{\frac{\sum_1^N (C_m - C_o)^2}{N}}}{\overline{C_o}}$$

(2)

where $C_m$ is the CMAQ-modeled value, $C_o$ is the observational value, $N$ is the number of simulation-observation pairs used in NMB and NRMSE calculations.

### 2.3 Aerosol pH calculation

In this study, we use the ISORROPIA-II thermodynamic model (Fountoukis and Nenes, 2007) to determine the composition in a $K^+$-$Ca^{2+}$-$Mg^{2+}$-$NH_4^+$-$Na^+$-$SO_4^{2-}$-$NO_3^-$-$Cl^-$-$H_2O$ aerosol system under equilibrium condition with gas phase precursors. Aerosol pH is calculated based on $H^+_{air}$ and liquid water content uptake by inorganic species ($LWC_i$) from ISORROPIA-II output:

$$pH = -log_{10} \gamma_{H^+} H^+_{aq} = -log_{10} \frac{1000 \gamma_{H^+} H^+_{air}}{LWC_i + LWC_o} \cong -log_{10} \frac{1000 \gamma_{H^+} H^+_{air}}{LWC_i}$$

(3)

where $\gamma_{H^+}$ is the activity coefficient of hydronium ion and is assumed to be 1 in this study (note that, the binary activity coefficients of ionic pairs, including $H^+$, are calculated in ISORROPIA-II), $H^+_{aq}$ ($mol \cdot L^{-1}$) is the hydronium ion concentration in the aerosol liquid water, $H^+_{air}$ ($\mu g \cdot m^{-3}$) is the equilibrium particle hydronium ion concentration per volume air. LWC includes $LWC_i$ and $LWC_o$, which means the water uptake by inorganic species and organic species ($\mu g \cdot m^{-3}$) are modeled separately because both organic and inorganic species are hygroscopic. In this study, we only consider the effect of $LWC_i$ since the effect of $LWC_o$ on aerosol pH has been found to be minor (Guo et al., 2015).

There are two modes in ISORROPIA-II's calculation, i.e. forward mode and reverse mode. In the forward mode, the inputs include total concentrations (i.e. gas+aerosol) of $TNH_3$, $TNO_3$, TCl ($HCl+Cl^-$), $SO_4$ and NVCs and meteorological parameters (temperature, relative humidity); in the reverse mode, only the aerosol phase of compounds and meteorological parameters are needed (Fountoukis and Nenes, 2007). In this study, the ISORROPIA-II model is run in the forward mode for aerosol in


metastable state because reverse mode has been reported to be more sensitive to measurement errors (Hennigan et al., 2015;Song et al., 2018).

We also find that there are measurements with unrealistically high $Ca^{2+}$ concentrations (such that $Ca^{2+}$ is more than LWC×0.002, i.e., the solubility of $Ca^{2+}$ in aerosol liquid water). This may be due to the measurement method of $Ca^{2+}$ which needs to use large amount of water to dissolve filter-based particles. This process will likely dissolve the water-insoluble part of $Ca^{2+}$ in

aerosols which may cause higher bias of aerosol $Ca^{2+}$ concentration. In the existence of aerosol $SO_4^{2-}$, $Ca^{2+}$ precipitates along with $SO_4^{2-}$ as $CaSO_4$ because of the low solubility (Seinfeld and Pandis, 2006). Including the high $Ca^{2+}$ concentration or not causes a large difference in predicted pH because of high acidity of $SO_4^{2-}$ (Sect. 3.2.4). In order to avoid the bias, we use modified $Ca^{2+}$ concentration for pH calculation, that is, we use original $Ca^{2+}$ concentration to calculate aerosol LWC and use the concentration of $Ca^{2+}$ that can dissolve in the LWC as the modified $Ca^{2+}$ concentration if original $Ca^{2+}$ is in excess of   its

solubility in the calculated LWC.

We evaluate the model performance by comparing the gas-particle partitioning of semi-volatile compounds between measured and simulated values such as $\varepsilon(NO_3^-)$ and $\varepsilon(NH_4^+)$. This method is effective when the species have substantial fractions in both gas and particle phases (Guo et al., 2017a). The comparison results of $\varepsilon(NH_4^+)$ and $\varepsilon(NO_3^-)$ are shown in Fig. S3. The correlation coefficients and the slopes of linear regression are all close to 1, suggesting good agreement between the simulations

and observations. In terms of these partitioning ratios, the model performs better in the United States than in China, which may attribute partly to the more even partitioning of the species between gas and particle phase in the United States.

**2.4 Multivariable Taylor Series Method (MSTM)**

In order to separate the contribution of each component (8 species in total, include $Na^+$, $SO_4$, $TNO_3$, $TNH_3$, $TCl$, $Ca^{2+}$, $K^+$, and $Mg^{2+}$) to the pH difference between China and the United States, we use a multivariable Taylor Series Method (MTSM). First,

we derive the average conditions (i.e., species concentrations and meteorological conditions) across all the sites in the United States and China. We then use the United States as the starting point and China as the end point and decompose the contributions of individual compounds to the pH difference based on the following equations:

$\Delta c_i = c_{i,China} - c_{i,US}$

(4)

$c_{i,\lambda} \cong c_{i,US} + \Delta c_i \cdot \lambda$

(5)

$\Delta pH = pH_{US} - pH_{China} = \int_0^1 \left( \sum_{i=1}^8 \frac{\partial pH}{\partial c_{i,\lambda}} \cdot \Delta c_i \right) \cdot d\lambda \cong \sum_{s=1}^{100} \left( \sum_{i=1}^8 \frac{\partial pH}{\partial c_{i,\frac{s}{100}}} \cdot \Delta c_i \right) \cdot 0.01$

(6)



$$\Delta pH_i \cong \sum_{s=1}^{100} \frac{\partial pH}{\partial c_{i,\frac{s}{100}}} \cdot \Delta c_i \cdot 0.01$$

(7)

where subscript $i$ denotes a specific species; $c_{i,China}$ and $c_{i,US}$ represent the concentration of compound $i$ in China and the United

States, respectively; $\Delta c_i$ is the difference in $c_i$ between China and the United States; $c_{i,\lambda}$ is an intervening $c_i$ between $c_{i,China}$ and

$c_{i,US}$ defined by $\lambda \in$ [0, 1]; $c_{i,\lambda}$ is $c_{i,US}$ when $\lambda$ is 0; $c_{i,\lambda}$ is $c_{i,China}$ when $\lambda$ is 1. In this study we assume negligible interaction

between species, therefore the increasing concentration of species i will not have the effect of changing the concentration of

other species. The pH difference between China and the United States (i.e., $\Delta pH$) can be expressed as the sum of the partial

derivatives of pH with respect to $c_{i,\lambda}$ which is then integrated from $c_{i,US}$ to $c_{i,China}$ , as described by Eq. (6). In this study, we

take 100 steps with equal intervals to gradually change $\lambda$ from 0 to 1 (Eq. (6)) and record the partial derivatives of pH with

respect to individual $c_{i,\lambda}$, and derive the contributions of all the species and meteorological variables to the pH change at every

step. By summing up the contributions at all the steps, we characterize the contributions of individual components to the overall

pH difference (Eq. (7)).

**3 Result and discussion**

**3.1 The pH difference between China and the United States**

**3.1.1 The pH difference based on observation**

The aerosol pH values calculated based on observational data show a significant difference between China (most observation

sites in NCP) and the United States. In China, the 2017 annual average pH level is 4.3 and ranges from 3.3 to 5.4 by monitoring

sites with an interquartile range of 3.9–4.6. In the United States, the 2011 annual average pH level is 2.6, ranging from 1.9 to

3.9 with an inter-quartile range of 2.2–3.0 (Fig. 1). The t-test shows a significant difference between the two groups (p<0.0001),

suggesting that the aerosols are on average more acidic at the monitoring sites in the United States than in China.

The pH difference can also be illustrated by the cumulative distribution function (CDF) curves (Fig. 2, solid lines). The shapes

of the CDF curves are similar in these two countries with a slightly steeper slope in the United States (Fig. 2a). The pH values,

however, are 1–2 units higher in China than in the United States at across levels of cumulative frequencies. In some cases, the

aerosol acidities could be completely neutral in China (the frequency is 2% for pH ≥7), while in the United States, the pH

values in all the cases are below 6.

Spatially, 14 out of the 16 sampling sites in China are in the NCP (Fig. S2c) which is one of the most populous and polluted

regions in China (Hu et al., 2014;Cui et al., 2020). Our pH results in this region are consistent with other studies (ranging from

3.5 to 4.6) (Liu et al., 2017;Ding et al., 2019;Ge et al., 2019). The distribution of sampling sites in the United States, on the

other hand, is evenly distributed spatially. The pH values in the Midwest and California are higher than in other regions like





the Southeast, in line with previous studies (Lawal et al., 2018;Chen et al., 2019). Overall, the pH level in the United States is 1.7 units lower than in NCP of China.

### 3.1.2 The pH difference based on model simulation

To solve the uneven spatial coverage of observational data in China, we conduct simulations using CMAQ, in company with the observational data, to further address the pH difference on a nation-wide scale. We evaluate the model performance by comparing the modeled and observed aerosol pH values (Fig. 3), major aerosol and gaseous species including $SO_4^{2-}$, $NO_3^-$, $NH_4^+$ and $HNO_3$, $NH_3$, and the partitioning ratios including $\varepsilon(NH_4^+)$ and $\varepsilon(NO_3^-)$, at monitoring sites (Fig. S4-S5).

Spatially, the model simulations generally capture the observed variations in pH, species concentrations, and partitioning ratios, although some biases occurred. For $SO_4^{2-}$, the model captures the high concentration in the NCP and the eastern US, but it shows low biases in some sites in the southern NCP. This leads to a more negative NMB of the modelled $SO_4^{2-}$ in China than in the United States, which can also be seen from Fig S5a& Fig S5i. Low biases are also found for other aerosol components, including $NH_4^+$, $NH_3$ and $NO_3^-$, in both countries. Such low biases have been seen in previous studies (Fountoukis et al., 2013;Theobald et al., 2016), which can be attributed to the spatial mismatch between the observations and simulations due to the coarse resolutions of the model grid cells (usually in the range of 20–50 km resolutions) (Shen et al., 2014;Wang et al., 2014). For $\varepsilon(NO_3^-)$, the model performs generally well, with high $\varepsilon(NO_3^-)$ in China and low $\varepsilon(NO_3^-)$ in the United States (Fig. S4f&Fig. S4vi, Fig. S5f& Fig. S5vi), although $\varepsilon(NO_3^-)$ in both countries are biased low (NMB is -28% in China and -30% in the United States) due to the lower bias in $NO_3^-$ than in $HNO_3$. For $\varepsilon(NH_4^+)$, the model reproduces the low levels in NCP and the high levels in the northeastern United States, but in China, $\varepsilon(NH_4^+)$ levels are biased high (NMB=52%) because of the low bias in $NH_3$ (Fig. S4g&Fig. S4e, Fig.S5g&Fig.S5e). The pH values predicted by the model are reasonable justified because 90% of the cases have the absolute pH differences between observation and simulation smaller than 1.2 in China and 2.0 in the United States. Both the NMBs and NRMSEs for pH are smaller in China than in the United States (Fig. 3).

With respect to the temporal variation, the model captures the seasonal trends of pH, $\varepsilon(NH_4^+)$, and $\varepsilon(NO_3^-)$ in both countries, all of which are lower in summer and higher in winter (Fig. 4). The lower temperature in wintertime favors the particle-phase for semi-volatile species. Comparison of the seasonal trends of the individual aerosol components shows a better agreement in the United States than in China. For example, the simulation in China misses the peaks of $SO_4^{2-}$ in winter and $NH_3$ in summer, and biases for $HNO_3$ in summer (Fig. S6a, i, e). On the other hand, the simulation in the United States captures the trends of almost all the components though is biased low for $SO_4^{2-}$ and $NH_4^+$ in summer (Fig. S6b, h). These results indicate the need for better quantification of the monthly emission trends in China which are currently subject to high uncertainty. Overall, the spatial and temporal evaluation suggests generally good agreement between the model simulations and observations in both countries.



In line with the pH comparison based on observational data (Sect.3.1.1), the nationwide model simulations show significant

differences in aerosol acidity between the two countries. Almost all the areas in the United States have aerosol pH values lower

than 3 according to the CDF plot (Fig. 2b). Higher pH values are found in the middle and eastern United States, while in the

western United States except California, the pH values are lower (Fig. 3). In China, a large portion of areas (87%) have aerosol

pH values above 3 according to the CDF plot, which is especially true in the eastern China with the largest population (Fig.

3). Aerosol pH values in the western and southeastern China are generally lower than in the east. The nationwide annual

average pH values in China and the United States are 2.7±0.6 and 0.8±0.8, respectively, lower than the observation-based

values because most of the monitoring sites are in the high pH areas (Fig. 3) and the bias in model simulation (Fig. 4a, Fig 4b).

Given the adverse health impacts of ambient aerosols (Burnett et al., 2014;Freedman et al., 2019) and the potential linkage of

aerosol acidity with aerosol toxicity through the solubility of redox-active metals (Oakes et al., 2012;Fang et al., 2015;Ye et

al., 2018), we further calculate and compare the population-weighted averages of the aerosol pH in the two countries in order

to highlight the pH levels in densely populated areas. The calculation shows the weighted pH values of 3.3±0.4 and 2.2±0.5 in

China and the United States, respectively, both of which are higher than non-weighted averages, which means that aerosols in

more populous areas tend to be less acidic (Fig. 2b). This finding is further confirmed by the significant positive correlation,

within each country, between the aerosol pH and population density (China: r=0.42, p<0.0001; the United States: r=0.28,

p<0.0001). Consistent with the observation-based results, the t-test for the model simulations shows a significant difference in

either the population-weighted or non-weighted aerosol pH values between the two countries (p< 0.001).

**3.2 Potential causes and effects of aerosol pH differences**

**3.2.1 Gaseous and aerosol compound profiles between China and the United States**

We further investigate the factors leading to the pH difference. Although both observations and simulations are subject to

uncertainty, we expect that the observational data should provide more direct and reliable evidence for this investigation, when

available. Table 1 summarizes the annual average concentrations of gaseous and aerosol species measured in China and the

United States during the study period (China: 2017, the United States: 2011). For all the gaseous and ionic species (except

$HNO_3$), the average concentrations in China are statistically significantly higher than those in the United States. The total

concentrations of WSI species in China (34.4 $\mu g \cdot m^{-3}$) are on average six times the concentrations in the United States (5.7

$\mu g \cdot m^{-3}$) and present a larger variation, ranging from 0.2–240 $\mu g \cdot m^{-3}$, compared to the range of 0.1–31 $\mu g \cdot m^{-3}$ in the United

States. Similar to other studies in China (Yao et al., 2002;Pathak et al., 2009;Zhang et al., 2013;Liu et al., 2016) and the United

States (Guo et al., 2015;Feng et al., 2020), $NH_4^+$, $NO_3^-$ and $SO_4^{2-}$, contribute more than 80% of the total WSI concentrations

in both countries. The mass fractions of individual WSIs, however, differ between the two countries (Fig. 5). In China, the

dominant WSI was $NO_3^-$ (34.6%), followed by $SO_4^{2-}$ (26.3%) and $NH_4^+$ (25.5%). In the United States in 2011, $SO_4^{2-}$ contributed

nearly half of the total WSI concentration (49.4%), and the contributions of $NO_3^-$ and $NH_4^+$ are comparable ($NO_3^-$ 17.6%, $NH_4^+$





18.8%), though $SO_4^{2-}$ and $NO_3^-$ levels have decreased dramatically along the years, leading to decreases in $NH_4^+$ due to less

substrate to interact with $NH_3$ and form particulate ammonium species (Butler et al., 2016).

As two of the most predominant anions in aerosols, the concentrations of $SO_4^{2-}$ and $NO_3^-$ at the monitoring sites in China are

4 and 15 times the concentrations in the United States, respectively. In particular, the relative difference in $NO_3^-$ between the

two countries is the most significant, compared with the differences in other WSI components. Hence the difference of the

nitrate to sulfate molar ratio ($NO_3^-/SO_4^{2-}$) is also significant in two countries. The observational data show that the ratios at

most monitoring sites in China are larger than 1, and that only two sites have the ratios lower but close to 1 (0.81, 0.94); on

the other hand, 27 out of 34 sites in the United States found a ratio lower than 1, ranging from 0.25–0.99, which are generally

lower than in China. High nitrate to sulfate ratio in China could be caused by more efficient oxidation of $NO_x$ than $SO_2$ in

China to allow larger amount of nitrate formation as well as higher aerosol pH and availability of ammonia which favor the

formation process of particle nitrate. The varying ratios of $NO_3^-/SO_4^{2-}$ on aerosol could further affect aerosol liquid water

uptake, which will be discussed in Sect. 3.2.4.

The most abundant cation in aerosols is $NH_4^+$, and the concentration difference of $NH_4^+$ between two countries is significant

compared with the difference of other cations. The average $NH_4^+$ level at the monitoring sites in China is more than ten times

the level in the United States. In addition, $\varepsilon(NH_4^+)$ in China (0.13–0.48) is approximately 50% lower than in the United States

(0.22-0.85), which means that compared to the United States, $TNH_3$ in China tends to present more in the gas phase. Higher

$NH_4^+$ and lower $\varepsilon(NH_4^+)$ levels in China indicate a higher level of $TNH_3$, which plays an important role on aerosol pH,

partitioning of $TNO_3$ and even particulate mass, discussed in Sect 3.2.3.

NVCs such as $Na^+$, $Ca^{2+}$, $Mg^{2+}$, and $K^+$ are often minor components of particles but important because of their ability to

neutralize acidic species in the atmosphere, such as sulfuric and nitric acids (Zhang et al., 2007). Neglecting NVCs makes low

biases of pH, driving the $NH_3$-$NH_4^+$ equilibrium shifting to the particle phase because more ammonium is used to neutralize

the aerosol acidity than should otherwise be neutralized by NVCs (Guo et al., 2018). Therefore, NVCs are included in

calculating aerosol pH in this study. High NVC concentrations usually occur at the sites near emission sources. For example,

high concentrations of $Na^+$, mainly from sea salt (Zhang et al., 2011), occur at Site 13,27, and 30 in the United States, which

are all coastal sites. The concentrations of $Ca^{2+}$, mainly from mineral dust, are found in greater abundance at Site 6, 11, 23 in

the United States and at Site 5 in China, which are prairies impacted by sand and dust. Average NVC concentrations in China

are up to an order of magnitude higher than in the United States, although in both countries, most of the NVCs concentrations

are small compared to $SO_4^{2-}$, $NO_3^-$, and $NH_4^+$. The predominant NVCs in China are $Ca^{2+}$ (2.8%), $Na^+$ (2.0%) and $K^+$ (2.1%),

while in the United States are $Ca^{2+}$ (5.9%) and $Na^+$ (3.7%).





### 3.2.2 Characterization of contribution to aerosol acidity by each component

We use MTSM as described in Sect. 2.4 to characterize the contribution of each component to the pH difference between the

United States and China. Three groups (i.e., observation, simulation non-weighted, simulation population-weighted) of the

annual average concentrations in the United States and China listed in Table S4 are chosen as the starting (the United States)

and ending (China) points and the results are shown in Fig. 6.

The average concentrations based on the observational and simulated data are not completely consistent due to the

representativeness of the monitoring sites and the discrepancy between the model simulations and observations. The MTSM

analyses based on the three groups, however, showed similar results, such as that all suggest the high $TNH_3$ in China as an

important factor leading to the difference in aerosol pH between the two countries (Fig. 6). The contribution of $TNH_3$ is the

highest in the "observation" group due to the large difference in $TNH_3$ concentration. The effects of other NVCs like $Na^+$, $K^+$,

$Mg^{2+}$ and $Ca^{2+}$ on the pH difference are also considerable. The NVCs in aggregate show contributions of 0.8, 0.6, 1.0 in the

"observation", "simulation", and "population-weighted simulation" groups, respectively, suggesting that the difference in

NVCs explains approximately 1-unit difference in aerosol pH between the two countries. The higher NVC contribution in the

"weighted simulation" group than in the "non-weighted" group may be explained by the interactions between NVCs and

anthropogenic emissions which have been found to result in a larger control of NVCs over aerosol pH (Guo et al., 2018;Wong

et al., 2020).

Unlike $TNH_3$ and NVCs which lead to higher pH values in China than in the United States, $SO_4^{2-}$ contributes oppositely to the

pH difference between the two countries. High $SO_4^{2-}$ concentrations decrease aerosol pH in China by 0.6–1.3 units (varying

by group), compared to the United States, although this effect is fully offset by $TNH_3$ and NVCs.

Compared to other species, the concentrations of $TNO_3$ are the most different between the two countries, but MTSM shows

that the contribution of $TNO_3$ on the pH difference is small. This result is further confirmed by a sensitivity test of $TNO_3$ (Fig.

9) which shows that the change in pH is subtle in two countries with the change in $TNO_3$ only.

### 3.2.3 Effects of ammonium on aerosol pH

The result of MTSM indicates that the difference in $TNH_3$ is one of the predominant reasons causing the pH difference. In

order to study the effect of $TNH_3$, we conduct sensitivity tests for China and the United States separately to investigate the

responses of aerosol pH to changing $TNH_3$. We change the $TNH_3$ concentrations from 0.1 to 1000 $\mu g \cdot m^{-3}$ while keep all other

components constant at their annual average levels based on observation data (Table 2). The results are shown in Fig. 7. It is

clearly illustrated that, over a large range of $TNH_3$ concentrations, aerosol pH increases with the increase in $TNH_3$ because the

production process of $NH_4^+$ from $NH_3$ consumes aqueous $H^+$. However, in both countries, aerosol pH has a small decrease

with the increase in $TNH_3$ when $TNH_3$ concentration is very low, this could be due to higher biases in $H^+$ concentration by

ISORROPIA in ammonia poor conditions (Ansari and Pandis, 1999). The local sensitivity of pH to $TNH_3$, expressed as the





pH increase per tenfold increase in $TNH_3$ at current $TNH_3$ level, is higher in the United States (3.0) than in China (0.4),

indicated a higher sensitivity of aerosol pH to $TNH_3$ in the United States than in China. Besides, we find that the responses of

pH to $TNH_3$ are nonlinear and anisotropic. With all others equal, pH in the United States could be closer to the level in China

if the $TNH_3$ increases to the level in China. On the other hand, the pH in China would be lower than the United States if the

$TNH_3$ decreases to the United States level because of the relative higher abundances of acidic components ($SO_4$, $TNO_3$, $TCl$)

than basic ions ($TNH_3$, NVCs) (Fig. 7a). In both countries, the sensitivities would quickly diverge from the original values

toward higher values as $TNH_3$ decreases, with the sensitivities in China changing at a faster pace. As $TNH_3$ increases, however,

the sensitivities in these two countries would gradually become constant, stabilizing at comparable levels (0.002 pH unit per

$TNH_3$ increase in both two countries).

The effects of $TNH_3$ on the gas-particle partitioning of $NH_3$-$NH_4^+$ and $HNO_3$-$NO_3^-$ are illustrated in Fig. 7b and 7c, showing a

decreasing trend of $\varepsilon(NH_4^+)$ and an increasing trend of $\varepsilon(NO_3^-)$ as $TNH_3$ increases. In the range of observation cases the value

of $\varepsilon(NH_4^+)$ in China is smaller than in the United States, suggesting excess presence of $TNH_3$ compared to other aerosol

components (e.g., $TNO_3$ and $SO_4$). $\varepsilon(NO_3^-)$ increased with increased $TNH_3$, due to higher aerosol pH which promote $TNO_3$

shifting to the particle phase as well as increased $NH_4^+$ promote the condensation of $HNO_3$ to form $NH_4NO_3$. Higher $\varepsilon(NO_3^-)$

in China than in the United States with an average $\varepsilon(NO_3^-)$ in China being close to 1 confirmed the excess presence of $TNH_3$.

Both the lower $\varepsilon(NH_4^+)$ and higher $\varepsilon(NO_3^-)$ in China estimated by the sensitivity curves are consistent with observations.

The gas to particle partitioning of $NH_3$ produces inorganic ammonium salt of ammonium bisulfate ($NH_4HSO_4$) and ammonium

sulfate (($NH_4)_2SO_4$) first because the affinity of sulfuric acid for $NH_3$ is much larger than that of nitric and hydrochloric acid

for $NH_3$, especially when $TNH_3$ concentration is relatively low (Behera et al., 2013). The excess $TNH_3$ may also react with

nitric acid and hydrochloric acid to form salt of $NH_4NO_3$ and $NH_4Cl$ which will dissolve in the aerosol liquid water (Zhao et

al., 2016). Therefore, the ratio of $[NH_4^+]$ to different acid ions ($[SO_4^{2-}]$, $[NO_3^-]$, $[Cl^-]$) can be used to indicate the relative

abundance of ammonia. To further investigate the effects of $TNH_3$ on pH at different levels of abundance, we divide the

observation data into three groups based on neutralization condition of particle phase $NH_4^+$. Group A contains the observations

when $[NH_4^+] < 2 \times [SO_4^{2-}]$, when available $NH_4^+$ cannot completely balance aerosol $SO_4$. Group B consists of the data points

when $2 \times [SO_4^{2-}] < [NH_4^+] < 2 \times [SO_4^{2-}] + [NO_3^-] + [Cl^-]$ when most of the aerosol $TSO_4$ is in the form of $SO_4^{2-}$ and excess $[NH_4^+]$

is available to stabilize nitrate and chloride driving the gas phase to shift to the particle phase. Group C contains the data points

when $[NH_4^+] > 2 \times [SO_4^{2-}] + [NO_3^-] + [Cl^-]$, where available $NH_4^+$ is enough to balance particle phase anions. We then investigate

the sensitivities of pH to $TNH_3$ in these three groups for China and the United States separately by changing the input $TNH_3$

from a median variation range (i.e. 55% to 150%) in each group in the two countries, respectively, and keeping all other

components (i.e., concentrations and meteorological conditions) unchanged. Note that no data in the United States fall in Group





C, making up only two groups in the United States (i.e., Groups A and B). The results with average values of each group are

shown in Fig. 8.

The aerosol pH increases with the increases in $TNH_3$ in all groups, which consist with the result of the sensitivity test in Fig.

7, but the increasing rates (i.e., the sensitivities of pH to $TNH_3$) and the pH levels vary among different groups (Fig. 8a). In

China, Group C that represents aerosol systems with largest amount of excess $NH_4^+$ shows the highest pH levels and the flattest

slopes of pH with $TNH_3$, suggesting a relatively low sensitivity of pH to the change in $TNH_3$ when $TNH_3$ is abundant. Group

A that represents aerosol systems with insufficient $NH_4^+$, shows the lowest pH levels with the slopes slightly steeper than in

Group C. As $TNH_3$ decreases to 55%, the average pH in China in Group A can be as low as 2.3, closer to the pH level in the

United States, consist with the conclusion in sensitivity test using average value only (Fig. 7a). Group B can be regarded as an

intermediate group between Groups A and C. But the sensitivities of pH to $TNH_3$ changes in group B are the highest among

the three groups when reducing $TNH_3$, which could be due to the rapid increase in $\varepsilon(NH_4^+)$ in this group as $TNH_3$ decreases

(Fig. 8b), that leads to a faster loss of $NH_4^+$ (Zheng et al., 2019). Note that although the relative abundance of $NH_4^+$ in group

B is smaller than in group C, the transition from group B to group C due to $TNH_3$ increase does not always happen. Because

if $TNH_3$ increase in an aerosol system with $2\times[SO_4^{2-}] < [NH_4^+] < 2\times[SO_4^{2-}]+[NO_3^-]+[Cl^-]$, $[NH_4^+]$ would increase, and more

$TNO_3$ and TCl would shift into the particle phase, leading to the increase of WSI concentration. However, the average WSI

concentration in group B is $55.03\pm46.79$ $\mu g \cdot m^{-3}$ in China, significantly higher than that in group C in China ($31.60\pm20.29$

$\mu g \cdot m^{-3}$). LWC in group B ($22.90\pm7.38$ $\mu g \cdot m^{-3}$) is also higher than that in group C ($14.37\pm16.85$ $\mu g \cdot m^{-3}$). We find that most of

the cases in group B could be identified as highly polluted cases where large amount of $NH_4NO_3$ is formed and dissolves in

the aerosol water. As a result, despite the higher abundance of $NH_4^+$ in group B than group A, $\varepsilon(NH_4^+)$ in group B is the highest

among all the groups (Fig. 8b).

Throughout the observed cases, 85% in China are in Group C (i.e., aerosol systems with excess $NH_4^+$), and 55% in the United

States are in Group A (i.e., aerosol systems with insufficient $NH_4^+$). The higher sensitivity of pH to $TNH_3$ in group A than in

group C explains why the pH sensitivity to $TNH_3$ increases more significantly in the United States than in China as $TNH_3$

decreases (Fig. 7a). Overall, the positive sensitivity of pH to $TNH_3$ and the different dominant groups in these two countries

(Group C in China, Group A in the United States) suggest that the high abundance of $TNH_3$ in China increases the aerosol pH

and is one of the major reasons for the pH difference between the two countries.

**3.2.4 The relationship between sulfate/nitrate and aerosol pH**

Besides the effect of $TNH_3$ on aerosol pH discussed in Sec 3.2.3, other species, especially the acidic species which mainly

include $SO_4$ and $TNO_3$, could also affect aerosol pH because of their effects on $H^+_{air}$ concentration as well as on LWC (Ding

et al., 2019). This effect is investigated in a sensitivity test by changing $TNO_3$ or $SO_4$ concentration while keeping other inputs

constant as the average levels (Fig. 9). Similar to the MSTM results as shown in Fig. 6, elevated $SO_4$ significantly increases



aerosol pH by increasing $H^+_{air}$. On the other hand, elevated TNO$_3$ only slightly increases $H^+_{air}$, indicating a weaker acidity than

that of TSO$_4$, in line with the result in a previous study (Guo et al., 2017b). This is partially due to the semi-volatile property

of TNO$_3$ (Ding et al., 2019). Notably, even in China where $\varepsilon(NO_3^-)$ are mostly close to 1, the variation of aerosol pH with

TNO$_3$ (roughly equals to NO$_3^-$ in this case) is also subtle. Therefore, for two systems with different moles of SO$_4^{2-}$ and NO$_3^-$

neutralized by same moles of NH$_4^+$, the system with more SO$_4^{2-}$ will likely have a lower pH. This result indicates that higher

aerosol acidity is associated with higher availability of TSO$_4$ rather than TNO$_3$, which can be confirmed by observed data in

Fig. 10.

Compared to the difference in TNO$_3$/TSO$_4$, the difference in NO$_3^-$/SO$_4^{2-}$ molar ratio is more significant due to higher aerosol

pH and ammonium in China promotes TNO$_3$ shift in particle phase as NH$_4$NO$_3$, leading to a higher NO$_3^-$/SO$_4^{2-}$ molar ratio,

while low pH in the United States promotes TNO$_3$ stay in gas phase, leading to a lower NO$_3^-$/SO$_4^{2-}$ ratio. Based on observation

data, 74.5% of the cases in China have NO$_3^-$/SO$_4^{2-}$ molar ratio larger than one, while only 22.3% in the United States. The

different NO$_3^-$/SO$_4^{2-}$ ratios, as a result of the pH difference as well as TNO$_3$/SO$_4$ difference in two countries, could subsequently

affect other aerosol properties, such as aerosol water uptake ability, which is one of the important reasons causing haze events

in China during winter time (Xie et al., 2019;Wang et al., 2020b). Although nitrate aerosol and sulfate aerosol absorbs similar

amounts of water per mass (Fig. S7), heavy haze events in China are usually associated with increased LWC with enhanced

RH levels under nitrate-dominate condition (Wang et al., 2020b). In order to study this effect, we categorize the observation

data into a nitrate-rich group (Group N, where [NO$_3^-$]/[SO$_4^{2-}$] > 3) and a sulfate-rich group (Group S, where [NO$_3^-$]/[SO$_4^{2-}$] <

1) and compare these two groups under different RH conditions. The ratio 3 in group N is mentioned in lab studies and is a

more typical value of nitrate-rich conditions in field observations (Ge et al., 1998;Xie et al., 2019).

The results in Fig. 11 show that aerosol pH values in the same groups in China and the United States have similar responses

to the changes in RH. In both countries, as RH increases, the pH in group N decreases, and the pH in group S increases (Fig.

11a). Both the values and the increasing rate of LWC in group N is larger than in group S, suggesting a higher water uptake

ability in nitrate-rich condition, which is likely due to higher aerosol mass compared with group S as shown in Fig. 11f. The

nearly two times aerosol mass in group N as in group S indicates the co-condensation effect of nitrate aerosol and LWC (Guo

et al., 2017a), which suggests that NO$_3^-$ formed in aerosol leads to a higher LWC due to the increase in aerosol mass, while

higher LWC dilutes $H^+_{air}$ and increases pH, which is favorable for more HNO$_3$ shifting from gas phase to particle phase and

thus continually increases particle NO$_3^-$ concentration. This effect will reach a balance when most of the gas phase HNO$_3$ is in

the particle phase with enough NH$_4^+$, and, therefore, $\varepsilon(NO_3^-)$ is close to 100% in group N in the two countries (Fig. 11e).

Besides, water uptake by hygroscopic aerosols increases the aerosol surface area and volume, enhancing the hydrolysis of

N$_2$O$_5$ across particles and forming NO$_3^-$ (Tian et al., 2018;Wang et al., 2020b):



The condition in group N usually has a higher LWC and aerosol mass, due to the mutual promotion between LWC and particle

nitrate. And such a condition in group N occurs more often in China than in the United States, which is probably one of the

reasons leading to high particle concentrations on hazy days in China.

**4 Discussions and implications**

Based on extended ground-level measurements and regional air quality model simulations, we find significant differences in

aerosol pH between China and the United States. Aerosols in the United States are on average more acidic with pH generally

1–2 units lower than in China. We use two independent methods, i.e., the MTSM method and sensitivity tests, to identify the

key factors leading to the pH difference. These two methods consistently reveal the important role of $TNH_3$ in causing the pH

difference. The MTSM method further shows a significant contribution of NVCs on the pH difference, and the sensitivity tests

highlight the high nitrate/sulfate ratios as one of the important responses to the pH difference, and high nitrate aerosol in China

will further lead to higher aerosol water uptake, which may have other effects to aerosol conditions.

The nitrate/sulfate ratio depends on the emission ratio of $NO_x/SO_2$, the availability of cations due to the dependency of $\varepsilon(NO_3^-)$

on $TNH_3$ (Fig. 8c, Fig. 9c), and other factors such as the atmospheric oxidizing capacity. Further investigation into the total

emissions shows that the emission molar ratios of $[NO_x]/[SO_2]$ are close to 3:1 in both countries (2.92 In China in 2017 and

3.12 in the United States in 2011 when assuming the emission $NO_x$ is in the form of $NO_2$), indicating that the emission

difference is not the major factor leading to the nitrate/sulfate ratio difference. On the other hand, the emission molar ratio of

$[NH_3]/([NO_x]+2\times[SO_2])$ in China (0.75) is 1.6 times higher than that in the United States (0.46), which is consistent with the

measured high relative abundance of $TNH_3$ in China and confirms that high availability of cations (mainly $NH_4^+$ caused by

high $NH_3$ emission ) is one of the causes for the high nitrate/sulfate ratio in China.

Will the aerosols in China be as acidic as in the United States as emissions are further controlled without significant reductions

in $TNH_3$? Unlikely. Although both countries have been taking actions to cut down pollutant emissions (Pinder et al., 2007;Hand

et al., 2012;Zhang et al., 2019), the reduction rates of $NO_x$ and $SO_2$ emissions are quite different between the two countries

(Fig. 12). In the United States, the reduction rates of $NO_x$ and $SO_2$ emissions (mainly from mobile and power sectors) were

similar during the past two decades, while the emission of $NH_3$ (mainly from the agricultural sector) kept constant. The data

in the monitoring sites in the United States showed a decreasing $SO_4^{2-}$ concentration over the years due to the $SO_2$ emission

reduction, but the reduction of $NO_3^-$ is not obvious compared with $SO_4^{2-}$ (Fig. S8). Lower $SO_4^{2-}$ concentration could lead to a

higher aerosol pH in the United States, but this effect could be buffered by partitioning of $TNH_3$, leading to a lower aerosol

pH than expected (Weber et al., 2016). Overall, significant higher $SO_4^{2-}$ concentration compared with relative stable $NO_3^-$

concentration still led to nitrate to sulfate ratio smaller than one. This ratio, however, reached a value higher than 1 in 2015,

four years after the period of this study (2011). In China, on the other hand, $SO_2$ emission reduction rate has been higher than

$NO_x$ reduction rate especially after the year 2012 (Fig. 12), which could lead to a higher nitrate to sulfate ratio (Wang et al.,

2020b). Although we don't have yearly aerosol concentration data in China, the shift from sulfate dominant aerosol to nitrate

dominant aerosol has been already observed and reported by previous studies (Wang et al., 2011;Xie et al., 2019). We also

collected the $PM_{2.5}$, $SO_4^{2-}$, $NO_3^-$ and $NH_4^+$ concentrations in different years from many other studies in three major cities

(Beijing, Shanghai, Guangzhou), which illustrate the increasing trend of the $NO_3^-/SO_4^{2-}$ ratio along the years (Table S4). As

emissions of $SO_2$ and $NO_2$ are being controlled in China, $NH_3$ is becoming relatively more abundant, which is evident from

the historical emission trends (decreasing $SO_2$ and $NO_x$ emissions vs. steady $NH_3$ emission, (Zheng et al., 2018)), neutralizing

the aerosol. Hence, aerosol pH in China will unlikely decrease as further emission control of $NO_x$ and $SO_2$ is implemented.

Previous studies have suggested that low aerosol pH is associated with increased toxicity because of the increased dissolubility

of transition metals in aerosol LWC, which induce airway injury and inflammation through the production of reactive oxygen

species in vivo (Kim et al., 2015). The lower aerosol pH in the United States than in China implies that aerosols in the United

States may be more toxic than in China after being inhaled by humans. However, this implication should be interpreted with

caution because there are other known pathways through which particulate matter can harm the human body and the

mechanisms of how particulate matter affects health have not been completely understood (Armstrong et al., 2004). More

studies are needed to address the health outcomes associated with the disparity in aerosol pH between the two countries.

**Author contribution**

HS initiated the research project. HS ran the model. HS and BZ designed the experiments analysed results and wrote the

manuscript. YH, SX, ZX helped with data preparation. All co-authors commented on the paper.

**Competing interests**

The authors declare that they have no conflict of interest.

**Data availability**

The data presented in this manuscript and the observational data in China can be obtained from the corresponding author upon

request. The observational data in China can also be obtained from the Data-sharing platform by Comprehensive Observation

Network for Air Pollution in Beijing-Tianjin-Hebei and Its surrounding Areas (http://123.127.175.60:8765/siteui/index). The

observational data in the United States can be obtained from Clean Air Status and Trends Network (CASTNET)

(https://www.epa.gov/castnet) and Ammonia Monitoring Network (AMoN) (http://nadp.slh.wisc.edu/amon/)

**Acknowledgements**

This research is supported by the U.S. Environmental Protection Agency (EPA grant number R835880), the National Science Foundation (NSF SRN grant number 1444745), and partially funded by the National Air Pollution Prevention Joint Research Center of China (grant number DQGG0204). Its contents are solely the responsibility of the grantee and do not necessarily

represent the official views of the supporting agencies. Further, the US government does not endorse the purchase of any commercial products or services mentioned in the publication.

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

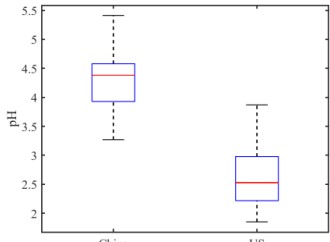

**Figure 1: Annual average aerosol pH at each monitoring site in China and the United States based on observational data.**

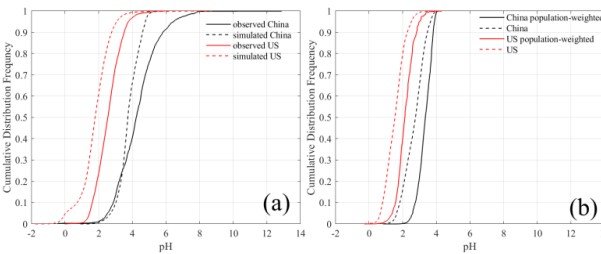

**Figure 2: The cumulative distribution function (CDF) curves of aerosol pH in China and the United States based on (a) observed particulate and gaseous composition (solid lines) and CMAQ simulations collocated with observation sites (dashed line); (b) simulated data nationwide.** In panel (b), both average and population weighted CDFs are shown.

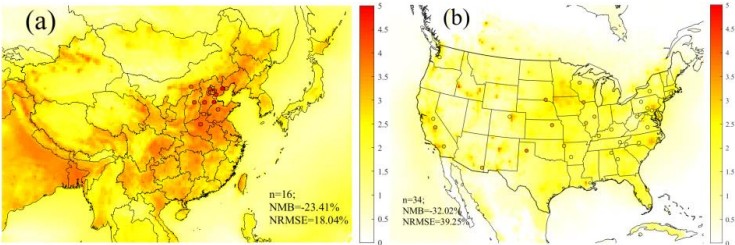

**Figure 3: Overlay of annual mean pH calculated based on simulated concentrations (colored map) and observed concentrations**
**(colored dots) over the study domain in (a) China and (b) the United States.** Number of sites(N), normalized mean bias (NMB) and normalized root-mean-square error (NRMSE) are provided in each figure.





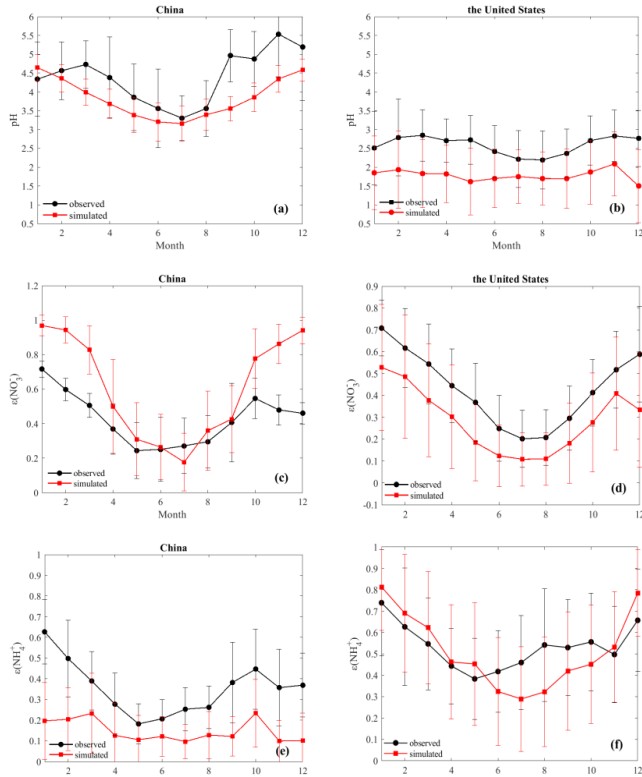

**Figure 4: Monthly average values of pH, ε(NO₃⁻) and ε(NH₄⁺) based on observed and CMAQ simulated data in China (a, c, e) and in the United States (b, d, f).**

China          The United States

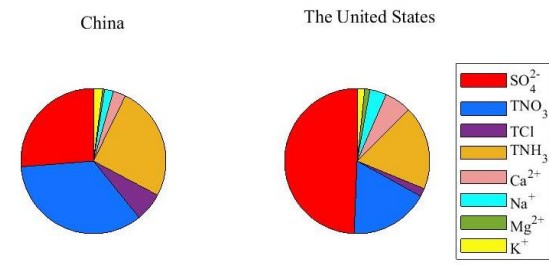


**Figure 5: Annual average values of water-soluble ions (WSI) concentrations profiles in China (left) and in the United States (right).**





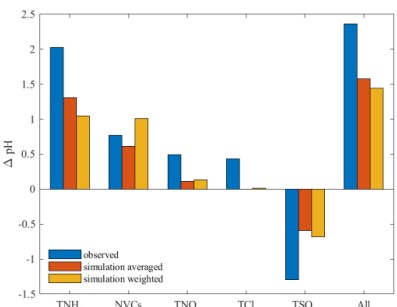

**Figure 6: Contribution of each compounds to aerosol pH difference between China and the United States calculated by multi-variable Taylor series method (MTSM) in Sect. 2.4.** The case in the United States is chosen as the starting point and China as the ending point.

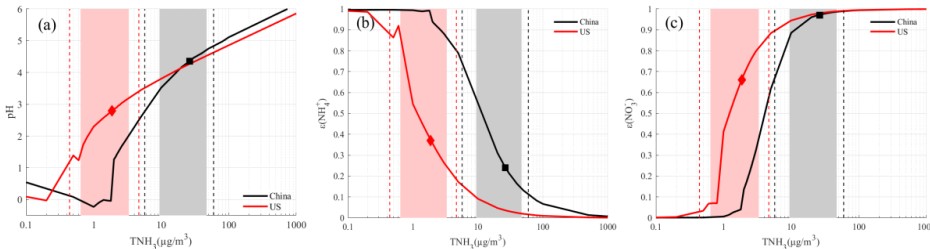

**Figure 7: Values of pH, ε(NH₄⁺) and ε(NO₃⁻) to the change of TNH₃ from 0.1 to 1000 μg·m⁻³ while keep all other components constant at their annual average levels.** The shaded areas show the TNH₃ concentration ranges that covers 75% of the observed cases in the countries, the black square and the red diamond mark the average TNH₃ levels in China and the United States, respectively.

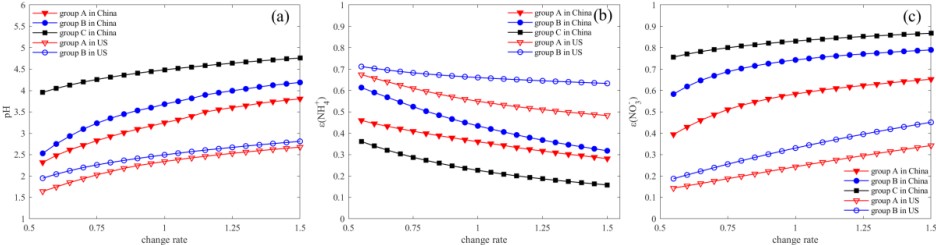

**Figure 8: Average values of pH, ε(NH₄⁺) and ε(NO₃⁻) when changing TNH₃ concentration from 55% to 150% in different groups.** Group A: China: pH=3.25±1.16, n=141; US: pH=2.41±0.72, n=651; Group B: China: pH=3.68±1.13, n=240; US: pH=2.75±0.69, n=540;Group C: China: pH=4.22±1.57, n=1312; US: n=0.



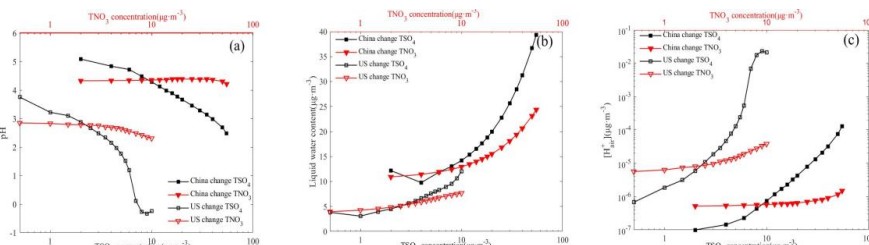

**Figure 9: Values of pH, liquid water content and H$^+_{air}$ to the change of TSO$_4$ and TNO$_3$ concentration in China and the United States.**

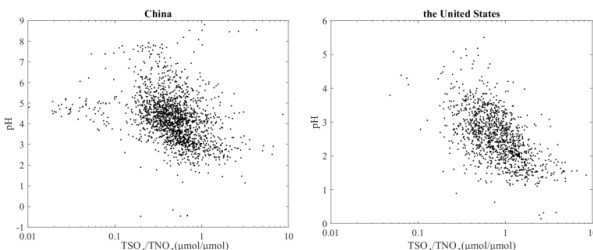

**Figure 10: The relation between aerosol pH and TSO$_4$/TNO$_3$ molar ratio in China (left) and the United States (right) based on observational data.**

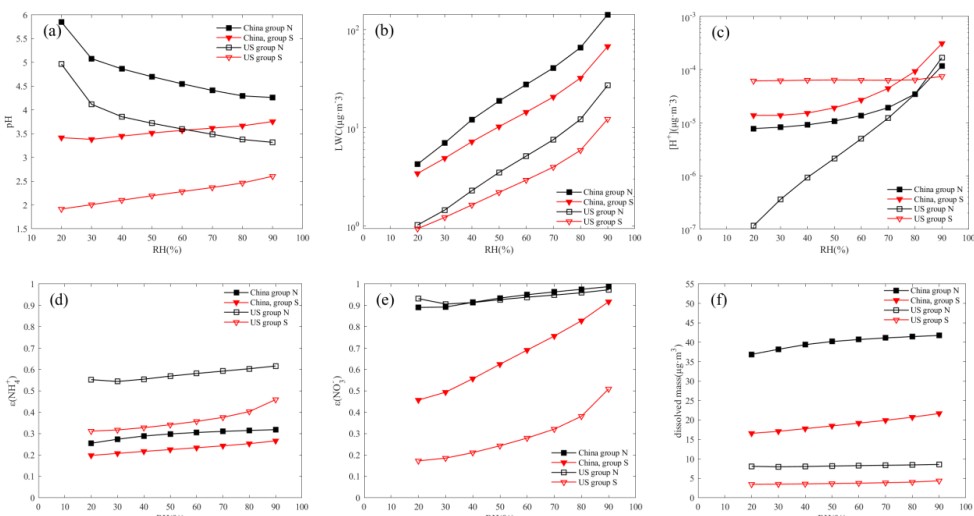

**Figure 11: Values of pH, LWC, H$^+_{air}$, ε(NH$_4^+$), ε(NO$_3^-$) and dissolved mass in group N and group S under different RH conditions in China and the United States.** China: group N, n=410; group S, n=470; US: group N, n=72; group S, n=1119.





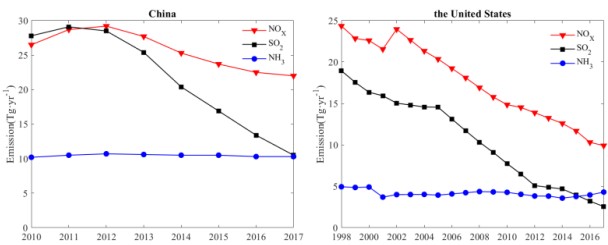

**Figure 12: Yearly trend of the emission of NH₃, NOₓ and SO₂ in China (left) and the United States (right).** The data in China are from studies by Zheng et al.(Zheng et al., 2018), the data in the United States are from Air Emissions Inventories by United States Environmental

Protection Agency (https://www.epa.gov/air-emissions-inventories/air-pollutant-emissions-trends-data)

**Table 1: Summary of the one-year average values of mass concentration of water soluble ions (WSI), gaseous and aerosol species, aerosol pH and meteorological parameters (as average ± standard deviation) in China and the United States during their study period (i.e. 2017 for China and 2011 for the United States).**

|  | China(n=1845) | US(n=1191) |
|---|---|---|
| **WSI($\mu$g·m$^{-3}$)** | 34.4±25.5 | 5.7±2.2 |
| **Temperature(K)** | 284.8±11.7 | 287.4±10.0 |
| **RH (%)** | 45.1±17.6 | 71.4±20.9 |
| **pH** | 4.3±1.2 | 2.6±0.7 |
| **Particle phase ($\mu$g·m$^{-3}$)** | | |
| **$SO_4^{2-}$** | 9.2±7.1 | 2.2±1.3 |
| **$NO_3^-$** | 12.1±11.1 | 0.8±0.9 |
| **$NH_4^+$** | 8.9±8.0 | 0.8±0.5 |
| **$Cl^-$** | 2.2±2.3 | 0.1±0.1 |
| **$Na^+$** | 0.7±1.0 | 0.2±0.2 |
| **$K^+$** | 0.7±0.6 | 0.1±0.1 |
| **$Ca^{2+}$** | 1.0±0.1 | 0.3±0.2 |
| **$Mg^{2+}$** | 0.2±0.1 | 0.1±0.1 |
| **Gaseous phase ($\mu$g·m$^{-3}$)** | | |
| **$NH_3$** | 18.0±12.6 | 1.1±1.7 |
| **HCl** | 1.9±3.4 | - |
| **$HNO_3$** | 1.0±1.1 | 1.0±0.6 |
| **Total ($\mu$g·m$^{-3}$)** | | |
| **$TNH_3$** | 26.5±17.2 | 1.9±1.8 |
| **TCl** | 4.1±4.5 | - |
| **$TNO_3$** | 13.1±11.2 | 1.8±1.1 |

745