# Peer review of "Significant contrasts in aerosol acidity between China and the United States"

_Atmospheric Chemistry and Physics, 2020_

## Referee Comment (RC1) · Anonymous Referee #1 · 18 Sep 2020

Review of "significant contrasts in aerosol acidity between China and the United States" by Zhang et al. NOTE: the title is actually "Unites" States, NOT "United" as I am sure the authors intended to write

The authors identify an important issue in atmospheric chemistry, namely the distribution in pH values across the globe and focus on 2 regions where SOx and NOx, two main contributors to aerosol acidity, are prevalent, the U.S. and China. The authors find that pH is generally higher in China than in the U.S. as a consequence of ammonia/ammonium and nitrate/nitric acid. Species focused on in this study, e.g., ammonium nitrate are volatile and are often not well described quantitatively in weekly (or longer) aggregated samples, as is characteristic of the U.S. samples used in this analysis. The authors point out that CASTNET's accuracy for most species, with the

exception of NH4+, is 'good'. I find this troubling because of the high time resolution measurements in China, to which the U.S. measurements are compared, and ammonium losses to the gas phase are a function of temperature, which changes over a week+ (U.S. measurements) and less so over and hour (China measurements). I find the lack of attention to the measurements hinders holistic interpretation of results.

For example, the authors point out that their model evaluation of partitioning ratios compares more favorably in the U.S. than in China and attribute this to "even more partitioning". They also state later in the manuscript: "On the other hand, the simulation in the United States captures the trends of almost all the components though is biased low for SO42- and NH4+in summer (Fig. S6b, h). These results indicate the need for better quantification of the monthly emission trends in China which are currently subject to high uncertainty." It is not immediately clear to me that this, in fact, means monthly emission trends in China are the driver. What about reasons for biases in the U.S.?

In the abstract the authors state: "Considering the historical emissions trends, the difference in aerosol acidity between these two countries is expected to continue as SO2 and NOx emissions are further controlled." If both countries are reducing emissions, it is not clear why this is the case when they do not provide context for this statement.

I cannot recommend publication of this manuscript in its current form.

detailed comments: Throughout the manuscript in the text and figures, the authors say "United States" and China, but more precisely mean the contiguous U.S. and Northern China Plains.

Page 15, Line 442: The authors state that emissions of NH3 in the U.S. have remained constant. Can they provide a reference? I do not think this is an accurate statement.

Does the midline in Figure 1 actually depict the average and not the median? Statistical software often defaults to the median.

Figure 4: what do the error bars represent?

Figures 10 and 12: It would be best to make the y-axis the same in each panel

There are several awkward English statement. I only list two: age 1, line 21: "adequate enough" page 8, line 226: 'reasonable justified"
* * *

---

## Referee Comment (RC2) · Anonymous Referee #2 · 15 Oct 2020

Review of "Significant contrasts in aerosol acidity between China and the Unites States" by B. Zhang et al.

This manuscript examines differences in aerosol pH between China and the US using both model simulations and network observations. The analysis investigates differences in aerosol pH between the two locations, primarily focusing on composition and concentration differences. Aerosol pH is an important topic and this work is timely and original. It is certainly appropriate for ACP and will be of interest to a broad scientific community. The organization is mostly fine, though some of the discussion is unnecessary (see comments below), and the writing is good. I do have some concerns that need to be addressed before I can recommend the manuscript for publication. My specific comments are below:

I think that the performance of the model is greatly overstated, as summarized in lines 235-237. As the manuscript states, model predictions of aerosol pH are frequently evaluated using comparison of modeled and measured species partitioning ($NH_3$ and $HNO_3$ are the most common species used). Figures S3, S4, and S5 show some significant problems predicting key species and parameters (especially $\varepsilon NH_4^+$ and $\varepsilon NO_3^-$), such that the pH predictions are also questionable in many times/locations. I think that these differences are mostly minimized in the manuscript, or not discussed accurately (e.g., Section 3.1.2). While some of the underlying differences are identified (e.g., the need for better NH3 emissions in China), the uncertainty in the pH predictions are not acknowledged. I think that the acceptable threshold for pH predictions should be much tighter than +/- 2 pH units (as line 225 – 228 seems to indicate). To address this concern, the manuscript needs to be more transparent and detailed in the discussion of the difficulties predicting both $\varepsilon NH_4^+$ and $\varepsilon NO_3^-$, and how this translates to uncertainty in the pH predictions.

This is no fault of the authors, but a significant paper was recently published that must be discussed (Zheng et al., Science 369, 1374–1377 (2020)), especially because the present manuscript presents several contrasting findings compared to Zheng et al. Specifically, Zheng et al. characteries differences in pH between China and the US, and the reasons for these changes. They find that the two most important factors are temperature and ALW. The present manuscript does account for ALW differences because composition and concentration both affect ALW; however, their analysis does not acknowledge the importance of temperature differences at all. Also, they discuss all of the differences as if composition has the biggest effect (e.g., adding NH3 neutralizes the acidic species…), when it may be the effect on ALW that is the most important factor, at least according to Zheng et al. Other studies have also identified the importance of temperature in driving pH differences (Battaglia et al., 2017; Tao and Murphy, 2019). Further, Zheng et al. conclude that NVCs make a very small contribution (on the order of ~5%) to the pH difference between the two regions, which seems to contradict the present study. So, the present manuscript needs to add significant discussion to address similarities and differences between their study and Zheng et al. They should also more broadly discuss other factors that are known to influence pH, such as temperature.

Finally, there is quite a bit of space (both figures and discussion) dedicated to analyses that don't seem to add much to the manuscript. For example, Line 350 describes the process for segregating the data into different groups to further examine the effects of TNH3. This was a good idea, however, the results (shown in Fig. 8) don't add any new insight to our current understand of aerosol pH. I would say the same is true for Fig. 11 and the associated discussion, and for the analysis of the TNO3/TSO4 molar ratios. I would strongly suggest moving these figures and discussion to the Supporting Information, especially in light of the added discussion and possible analyses needed to address the above comments.

**Technical Corrections**

Line 51: delete the period appearing in the middle of the sentence

Line 62-63: what are "large-scale" measurements?

Line 93: I question the stated accuracy of the AMoN NH3 measurements – especially given the variability between duplicate samples reported by the network.

Line 97: "Its" should not be capitalized

Line 101 – 103: provide the criteria for identifying outliers, and the number of outliers excluded from the respective datasets

Line 158-159: specify if sulfate was also adjusted?

Line 241-241: need to acknowledge that most of the pH predictions over China cannot be evaluated due to limitations in observational data.

Line 251-252: these correlations are weak, so the description of a "significant positive correlation" is misleading.

Line 325: "this could be due to higher biases in H+ concentration by ISORROPIA in ammonia poor conditions" – I don't follow this explanation?

Line 457: awkward as written

**References**

Battaglia, M. A., Douglas, S., and Hennigan, C. J.: Effect of the Urban Heat Island on Aerosol pH, Environ. Sci. Technol., 51, 13095–13103, https://doi.org/10.1021/acs.est.7b02786, 2017.

Tao, Y. and Murphy, J. G.: The sensitivity of PM$_{2.5}$ acidity to meteorological parameters and chemical composition changes: 10-year records from six Canadian monitoring sites, Atmos. Chem. Phys., 19, 9309–9320, https://doi.org/10.5194/acp-19-9309-2019, 2019.

Zheng, G., Su, H., Wang, S., Andreae, M.O., Pöschl, U., Cheng, Y., Multiphase buffer theory explains contrasts in atmospheric aerosol acidity, *Science*, 369 (1374-1377), 2020.

---

## Author Comment (AC1) · 17 Feb 2021

**Response to Anonymous Referee #1**
Manuscript: *Significant contrasts in aerosol acidity between China and the United States*
Manuscript number: acp-2020-879
Journal: Atmospheric Chemistry and Physics
Authors: Bingqing Zhang, Huizhong Shen, Pengfei Liu, Hongyu Guo, Yongtao Hu, Yilin Chen, Shaodong Xie, Ziyan Xi, T. Nash Skipper, Armistead G. Russell
* * *
**Comment 1**

The title is actually "Unites" States, NOT "United" as I am sure the authors intended to write.

**Response**

Thank you for pointing out this typo. "Unites" was changed to "United" in the title of the revised version.
* * *
**Comment 2**

The authors identify an important issue in atmospheric chemistry, namely the distribution in pH values across the globe and focus on 2 regions where SOx and NOx, two main contributors to aerosol acidity, are prevalent, the U.S. and China. the authors find that pH is generally higher in China than in the U.S. as a consequence of ammonia/ammonium and nitrate/nitric acid.

**Response**

We thank the reviewer for their review and the constructive comments. In the revised manuscript, we added the discussion about the impacts of the long duration of the CASTNET sampling approach on estimated pH and the potential reasons for the bias in modeled temporal variation. We clarified that the results based on observations in China are more representative of North China Plain to avoid misunderstanding and added more results and discussion on the nationwide model simulations. We hope that this new version of the manuscript addressed all the reviewer's concerns.
* * *
**Comment 3**

Species focused on in this study, e.g., ammonium nitrate are volatile and are often not well described quantitatively in weekly (or longer) aggregated samples, as is characteristic of the U.S. samples used in this analysis. The authors point out that CASTNET's accuracy for most species, with the exception of NH4+, is 'good'. I find this troubling because of the high time resolution measurements in China, to which the U.S. measurements are compared, and ammonium losses to the gas phase are a function of temperature, which changes over a week+ (U.S. measurements) and less so over and hour (China measurements). I find the lack of attention to the measurements hinders holistic interpretation of the results.

**Response**

We thank the reviewer for pointing out this issue regarding the long duration of the CASTNET sampling system. Through a literature search, we found that a previous study (Sickles et al., 1999) conducted a comprehensive comparison of the CASTNET weekly-duration sampling approach with a 24-h-duration sampling approach. Both approaches used filter packs. They found that compared to the 24-h duration, the weekly duration led to low biases of -5%, -5%, and -0.7%, on average, in measured $HNO_3$, $NO_3^-$, and $NH_4^+$, respectively, and high biases of 4% and 16%, on average, in $SO_4^{2-}$ and $SO_2$, respectively. In the revision, we conducted a sensitivity test that incorporated these reported biases associated with the long-duration of CASTNET sampling approach to adjust the calculated pH values in the United States. The sensitivity test suggested that the adjusted pH values showed little difference from the original ones ($2.69\pm0.85$ and $2.74\pm0.83$ on average for the original pH and the adjusted pH,

respectively). We added the description of this sensitivity test in Sect. 2.1 (observational data, lines 108-115) and the results of this test in Sect. 3.1.1 (the pH difference based on observations, lines 196-200) to point out the potential biases associated with the long duration of CASTNET samples.

The text added in Sect. 2.1 is as follows,

"*It should be noted that the weekly (or longer) duration of the CASTNET samples in the US may lead to biases in the measured concentrations especially for volatile species such as ammonium nitrate. Sickles et al. (1999) conducted a comprehensive comparison of measurements using the CASTNET weekly-duration sampling approach with those using a 24-h-duration sampling approach. Both approaches used filter packs. They found that compared to 24-h sampling, weekly sampling led to low biases of -5%, -5%, and -0.7%, on average, in measured $HNO_3$, $NO_3^-$, and $NH_4^+$, respectively, and high biases of 4% and 16%, on average, in $SO_4^{2-}$ and $SO_2$, respectively. To evaluate the potential biases in the calculated aerosol pH due to the weekly-duration sampling, we conduct a sensitivity test to adjust the CASTNET-measured concentrations based on the reported average differences between weekly-duration and 24-h-duration samples (Sickles et al., 1999) (Results and Discussion).*"

The text added in Sect. 3.1.1 is as follows,

"*The sensitivity test to adjust the CASTNET-measured concentrations based on the reported average differences between weekly-duration and 24-h-duration samples shows little difference between the unadjusted and adjusted pH values in the US (2.69±0.85 and 2.74±0.83 on average for the unadjusted and adjusted pH, respectively), suggesting that the weekly duration of the CASTNET sampling has little impact on the calculated aerosol pH. Therefore, we proceed with our subsequent analyses using the unadjusted pH.*"
* * *
**Comment 4**

For example, the authors point out that their model evaluation of partitioning ratios compares more favorably in the U.S. than in China and attribute this to "even more partitioning". They also state later in the manuscript: "On the other hand, the simulation in the United States captures the trends of almost all the components though is biased low for SO42- and NH4+in summer (Fig. S6b, h). These results indicate the need for better quantification of the monthly emission trends in China which are currently subject to high uncertainty." It is not immediately clear to me that this, in fact, means monthly emission trends in China are the driver. What about reasons for biases in the U.S.?

**Response**

We are sorry for the confusion. In this study, we conducted two comparisons. The first comparison was based solely on measurement data, whereby we compared the measured gas/particle partitioning ratios with the ratios re-partitioned by ISORROPIA-II using measured total (gas+particle) concentrations as inputs. This is a common approach to checking measurement data quality (Guo et al., 2016; Guo et al., 2017). The second comparison was to compare the measured concentrations with CMAQ-predicted concentrations. This comparison was used to evaluate the CMAQ model performance. The results of the first comparison were shown in Fig. S3, and the results of the second comparison in Fig. S4–6. The two statements mentioned by the reviewer, i.e., "more even partitioning" and "On the other hand, the simulation in the United States…", interpreted results of different comparisons, which could lead to confusion if they were thought to come out of the same comparison.

To avoid this confusion, the sentence in line 165 (line 161 in the original version), "We evaluate the

model performance by comparing the gas-particle partitioning of semi-volatile compounds between measured and simulated values such as ε(NO₃⁻) and ε(NH₄⁺)", was revised as "*We compare the directly measured gas-particle partitioning ratios of semi-volatile compounds with the ratios re-partitioned by ISORROPIA-II using measured total (gas+particle) concentrations as inputs. The purpose of this comparison, as conducted in previous studies (Guo et al., 2016; Guo et al., 2017), is to examine the measurement data quality.*"

The sentences in lines 169-172 (lines 163-166 in the original version) were revised as follows to further clarify that the statement, "more even partitioning", refers to the results from the first comparison (i.e., the comparison between measured and ISORROPIA-II-re-calculated partitioning ratios),

"*The correlation coefficients and the slopes of linear regression are all close to 1, suggesting good agreement between the measured and ISORROPIA-re-calculated partitioning ratios. In terms of these partitioning ratios, the model (ISORROPIA-II) performs better in the US than in China, which may be attributable, in part, to the more even partitioning of the species between gas and particle phase in the US.*"

In response to the second statement mentioned in this comment, the sentences in lines 247-252 (lines 232-235 in the original version) as revised as follows to provide possible reasons for the biases in the US,

"*For example, the simulation in the US captures the trends of almost all components, though it is biased high for $SO_4^{2-}$ and $NH_4^+$ in summer (Fig. S6 b and h); the simulation in China misses the peaks of $SO_4^{2-}$ in winter and $NH_3$ in summer, and has high biases for $HNO_3$ in summer (Fig. S6 a, i, and e). Measurement-related biases may contribute to the disparity in the temporal trends between observed and modeled concentrations. The uncertainty in monthly profiles of emission estimates may also play an important role. For example, CASTNET's long sampling period could lead to a larger measurement bias in summer than in winter (Sickles and Shadwick, 2008); the large uncertainty in the current estimates of $NH_3$ emissions in China, especially the reported underestimation of summertime emissions as indicated by an inversion analysis (Kong et al., 2019), may cause the absence of the summertime $NH_3$ peak in the simulated trend (Fig. S6i). Further investigation is needed to better understand the factors underpinning the disparity between observations and model simulations.*"
* * *
**Comment 5**

In the abstract the authors state: "Considering the historical emissions trends, the difference in aerosol acidity between these two countries is expected to continue as SO2 and NOx emissions are further controlled." If both countries are reducing emissions, it is not clear why this is the case when they do not provide context for this statement.

**Response**

Thank you for pointing this out. This sentence in the abstract as well as the discussion of the emission trend in Discussions and Implications (Section 4 in original version) has been removed in the updated manuscript.
* * *
**Comment 6**

Throughout the manuscript in the text and figures, the authors say "United States" and China, but more precisely mean the contiguous U.S. and Northern China Plains.

**Response**

We thank the reviewer for pointing this out. In this study, we compared the aerosol pH difference between these two countries based on multiple sources, including monitoring networks and model simulations. The monitoring network in China only covers Northern China Plains (NCP), as pointed out by the reviewer and clarified in multiple places in the manuscript. The model simulations, on the other hand, covers entire areas in China and the contiguous United States. In Sect. 3.1.2, we reported significant differences in aerosol acidity between these two countries even considering areas other than NCP in China. In Fig. 2(b), we derived the cumulative distribution function (CDF) based on model simulations that cover the entire China and the contiguous United States domains. We found that the cumulative frequency at the same pH level is always higher in China than in the contiguous United States, both with and without population as weight. In lines 260 and 266, we calculated the domain-wide average pH levels in these two countries, and the values are 2.7±0.6 in China and 0.8±0.8 in the contiguous United States without population as weight, and 3.3±0.4 in China and 2.2±0.5 in the contiguous United States with population as weighted.

In Sect. 3.2.2, we used Multivariable Taylor Series Method (MTSM) based on both observations and simulations to characterize the contribution of each component. The simulation data in this analysis again covered the entire areas in China and the contiguous United States. Analyses based on both observations and simulations (Fig. 6) consistently showed that $TNH_3$ and $SO_4^{2-}$ have the largest contribution to aerosol acidity difference while others have relatively small contribution.

In response to this comment, we further clarified in line 275 that observations in China were clustered in NCP as follows,

"*It should be noted that the monitoring sites in China were clustered in NCP and, thus, may not be representative of the whole of China.*"

We changed the term "China" to "*NCP*" or "*China (NCP)*" when interpreting the results based on observations in China. We also changed the term "the United States" or US to "*the contiguous United States*" or the contiguous US in proper places in the text. For example, the title of section 3.2.1 was revised as "*Gaseous and aerosol compound profiles between China (NCP) and the contiguous US*"; lines 277 was revised as "*...measured in China (NCP) and the contiguous US...*"; lines 279 was revised as "…concentrations in China (NCP)…"; lines 284 was revised as "In China (NCP)…" as well as multiple places elsewhere.

In the revised section 3.2.2 and the newly added section 3.2.3, we focused our interpretation on three groups. Two groups were derived from model simulations to ensure a nationwide coverage of our analyses. Please see these two sections for details.

In Supplementary Information, we added more analyses and discussion of the effect of $TNH_3$ based on nationwide simulation results (Text S1, Sect. 3.2.3 in the original version)

Fig. S10 (Fig. 7 in the original version) was revised to add the results based on simulations as follows,

[Figure]

**Fig.S10 Responses of pH, ε(NH$_4^+$) and ε(NO$_3^-$) to the change of TNH$_3$ from 0.1 to 1000 μg·m$^{-3}$ while keep all other components constant at their annual average levels.** The shaded areas show the TNH$_3$ concentration ranges that covers 75% of the observed cases in the countries, the dashed lines show the 5th and 95th percentiles of the observed cases, the black square and the red diamond mark the average TNH$_3$ levels in China and the United States, respectively.

**Comment 7**

Page 15, Line 442: The authors state that emissions of NH3 in the U.S. have remained constant. Can they provide a reference? I do not think this is an accurate statement.

**Response**

We derived this conclusion based on Figure 12 in original version, and the data is from the National Emission Inventory (NEI) released by United States Environmental Protection Agency (https://www.epa.gov/air-emissions-inventories/air-pollutant-emissions-trends-data). In a 14-year period from 1998 to 2011, the NH$_3$ emission changed from 4.94 Tg·yr$^{-1}$ to 4.03 Tg·yr$^{-1}$. The variation is much less than that of SO$_2$ and NO$_x$. The NEI document provided the emission data from a longer period (1990-2019), which shown in the following figure.

[Figure]

**Yearly trend of the emission of NH$_3$ in the United States**, the data in the United States are from Air Emissions Inventories by United States Environmental Protection Agency (https://www.epa.gov/air-emissions-inventories/air-pollutant-emissions-trends-data)

This figure and related discussion have been removed from the revised manuscript.

**Comment 8**

Does the midline in Figure 1 actually depict the average and not the median? Statistical software often defaults to the median.

**Response**

Thanks for pointing this out. Yes, the midlines in the original figure depict the median. We have removed the previous midlines and added the lines representing the averages.

For comparison, the figure changed from (a) to (b).

[Figure]

(a) median  (b) average

The following sentence was added to the end of the figure caption to clarify this,
"*The arithmetic mean (midline), the interquartile range (box), and the minimum-maximum range (whiskers) are shown in the box plot.*"

**Comment 9**

Figure 4: What do the error bars represent?

**Response**

The error bars represent the standard deviation of all the cases in each month, which indicate the variation among different sites in two countries. In response to this comment, we added the description "*The error bars represent the standard deviation of all the cases in each month*" in the caption of Fig. 4 and Fig. S6.

**Comments 10**

It would be best to make the y-axis the same in each panel

**Response:**

Thanks for the suggestion, we have made the y-axis the same in each panel as follows. This figure has been moved to SI as Fig. S12 in the revised manuscript.

[Figure]

**Comments 11**

There are several awkward English statements. I only list two: line 21:"adequate enough", page 8, line 226:"reasonable justified".

**Response**

In response to this comment, the language of the revised manuscript was checked by two native English speakers. Nash Skipper was added to the author list due to his contribution to the editing of the revised manuscript.

We thank the reviewer for their constructive comments and detailed suggestions. The quality of the manuscript has been substantially improved thanks to their review.

**Reference**

Guo, H., Sullivan, A. P., Campuzano-Jost, P., Schroder, J. C., Lopez-Hilfiker, F. D., Dibb, J. E., Jimenez, J. L., Thornton, J. A., Brown, S. S., Nenes, A., and Weber, R. J.: Fine particle pH and the partitioning of nitric acid during winter in the northeastern United States, Journal of Geophysical Research: Atmospheres, 121, 10,355-310,376, 10.1002/2016jd025311, 2016.

Guo, H., Liu, J., Froyd, K. D., Roberts, J. M., Veres, P. R., Hayes, P. L., Jimenez, J. L., Nenes, A., and Weber, R. J.: Fine particle pH and gas–particle phase partitioning of inorganic species in Pasadena, California, during the 2010 CalNex campaign, Atmos. Chem. Phys., 17, 5703-5719, 10.5194/acp-17-5703-2017, 2017.

Kong, L., Tang, X., Zhu, J., Wang, Z., Pan, Y., Wu, H., Wu, L., Wu, Q., He, Y., Tian, S., Xie, Y., Liu, Z., Sui, W., Han, L., and Carmichael, G.: Improved Inversion of Monthly Ammonia Emissions in China Based on the Chinese Ammonia Monitoring Network and Ensemble Kalman Filter, Environ Sci Technol, 53, 12529-12538, 10.1021/acs.est.9b02701, 2019.

Sickles, I. J. E., Hodson, L. L., and Vorburger, L. M.: Evaluation of the filter pack for long-duration sampling of ambient air, Atmospheric Environment, 33, 2187-2202, https://doi.org/10.1016/S1352-2310(98)00425-7, 1999.

Sickles, J. E., and Shadwick, D. S.: Comparison of particulate sulfate and nitrate at collocated CASTNET and IMPROVE sites in the eastern US, Atmospheric Environment, 42, 2062-2073, https://doi.org/10.1016/j.atmosenv.2007.11.051, 2008.

---

## Author Comment (AC2) · 17 Feb 2021

**Response to Anonymous Referee #2**
Manuscript: *Significant contrasts in aerosol acidity between China and the United States*
Manuscript number: acp-2020-879
Journal: Atmospheric Chemistry and Physics
Authors: Bingqing Zhang, Huizhong Shen, Pengfei Liu, Hongyu Guo, Yongtao Hu, Yilin Chen, Shaodong Xie, Ziyan Xi, T. Nash Skipper, Armistead G. Russell
* * *
**Comment 1**

This manuscript examines differences in aerosol pH between China and the US using both model simulations and network observations. The analysis investigates differences in aerosol pH between the two locations, primarily focusing on composition and concentration differences. Aerosol pH is an important topic and this work is timely and original. It is certainly appropriate for ACP and will be of interest to a broad scientific community. The organization is mostly fine, though some of the discussion is unnecessary (see comments below), and the writing is good. I do have some concerns that need to be addressed before I can recommend the manuscript for publication. My specific comments are below:

**Response**

We thank the reviewer for their review and the overall support. In the revised version, we carefully addressed all the reviewer's comments. We rewrote the model evaluation section to avoid overstatement and added more detailed discussion on model performance and results. As suggested by the reviewer, we also provided discussion about the seemingly contradicting conclusion compared to Zheng's study. The reviewer's suggestion on LWC enlightened us to identify two pathways through which factors affect aerosol pH—the LWC-modifying pathway and the $H^+$-modifying pathway. A new section (sect. 3.2.3) was thus added to provide related results and discussion. Some results and discussion in the original version of the manuscript have been moved to Supplementary Information per the reviewer's request. We hope that our revision and this new version of the manuscript have addressed all the reviewer's concerns.
* * *
**Comment 2**

I think that the performance of the model is greatly overstated, as summarized in lines 235-237. As the manuscript states, model predictions of aerosol pH are frequently evaluated using comparison of modeled and measured species partitioning (NH3 and HNO3 are the most common species used). Figures S3, S4, and S5 show some significant problems predicting key species and parameters (especially εNH4+ and εNO3-), such that the pH predictions are also questionable in many times/locations. I think that these differences are mostly minimized in the manuscript, or not discussed accurately (e.g., Section 3.1.2). While some of the underlying differences are identified (e.g., the need for better NH3 emissions in China), the uncertainty in the pH predictions are not acknowledged. I think that the acceptable threshold for pH predictions should be much tighter than +/- 2 pH units (as line 225 – 228 seems to indicate). To address this concern, the manuscript needs to be more transparent and detailed in the discussion of the difficulties predicting both εNH4+ and εNO3-, and how this translates to uncertainty in the pH predictions.

**Response**

We thank the reviewer for their suggestion on the discussion of the model performance. In response to this comment, we revised the model performance section (in lines 223-239) to avoid overstatement and add more transparent and detailed discussion of the difficulties in predicting $\varepsilon(NH_4^+)$ and $\varepsilon(NO_3^-)$ and how this translates to uncertainty in modeled pH in the model evaluation section as follows,

*"Spatially, the model simulations generally capture the observed variations in pH, species concentrations, and partitioning ratios, although there are some notable biases (Figs. S4 and S5). In both China (NCP) and the contiguous US, the modeled $NH_4^+$, $NO_3^-$, and $NH_3$ are biased low while modeled $HNO_3$ is biased high, resulting in low biases in the predicted $\varepsilon(NO_3^-)$ and $\varepsilon(NH_4^+)$. The modeled $SO_4^{2-}$ in both countries is biased low. Such low biases have been seen in previous studies (Fountoukis et al., 2013; Theobald et al., 2016) and have been attributed to the spatial mismatch between the observations and simulations due to the coarse resolutions of model grid cells (usually 20–50 km resolution) (Shen et al., 2014; Wang et al., 2014a). Smaller NMBs in the US indicate a better performance, compared to China (NCP). Larger differences between observations and simulations in China (NCP) could also be caused by larger measurement uncertainties as the data in China are collected from different monitoring stations operated by individual research institutions (Wang et al., 2019) and thus lack a unified quality control, compared with data in the US, which come from national monitoring networks (United States Environmental Protection Agency;National Atmospheric Deposition Program). The co-occurrence of low biases in $\varepsilon(NO_3^-)$, which causes lower bias in aerosol pH, and low biases in $\varepsilon(NH_4^+)$ and $SO_4^{2-}$, which cause higher bias in aerosol pH, likely offset each other, resulting in small biases in aerosol pH. Indeed, the simulated average pH values at observation sites (3.8±0.2 in NCP, China and 1.8±0.5 in the contiguous US) are generally in line with the observed averages (4.3±0.5 in NCP, China and 2.6±0.5 in the contiguous US) (Fig. 3), although the model shows a moderate low bias in both countries. The larger pH difference in the US than in China is likely due to the low bias in $TNH_3$ to which the sensitivity of pH is found to be more pronounced in the US than in China (discussed in detail in Text S1)."*

**Comment 3**

This is no fault of the authors, but a significant paper was recently published that must be discussed (Zheng et al., Science 369, 1374–1377 (2020)), especially because the present manuscript presents several contrasting findings compared to Zheng et al. Specifically, Zheng et al. characteries differences in pH between China and the US, and the reasons for these changes. They find that the two most important factors are temperature and ALW. The present manuscript does account for ALW differences because composition and concentration both affect ALW; however, their analysis does not acknowledge the importance of temperature differences at all. Also, they discuss all of the differences as if composition has the biggest effect (e.g., adding NH3 neutralizes the acidic species…), when it may be the effect on ALW that is the most important factor, at least according to Zheng et al. Other studies have also identified the importance of temperature in driving pH differences (Battaglia et al., 2017; Tao and Murphy, 2019). Further, Zheng et al. conclude that NVCs make a very small contribution (on the order of ~5%) to the pH difference between the two regions, which seems to contradict the present study. So, the present manuscript needs to add significant discussion to address similarities and differences between their study and Zheng et al. They should also more broadly discuss other factors that are known to influence pH, such as temperature.

**Response:**

Thank you for sharing this recently published Science paper (or Zheng's paper). Zheng's paper provided in-depth analysis of the drivers leading to the aerosol pH difference between northern China and the United States based on a multiphase buffer theory that they proposed. Their results as illustrated in Fig. 3 of their paper highlighted aerosol water content (AWC) and temperature as two most important factors explaining the pH difference, which seemingly contradicts our findings that mainly attribute the pH difference to $TNH_3$ concentrations and aerosol composition. First of all, we

want to thank the reviewer for raising this interesting question which should definitely be addressed in the current paper. Below we listed the reasons that explain the contrasting conclusions. Corresponding discussion was also reflected in the revised manuscript.

**Temperature**. Zheng's paper compared two scenarios with very distinct conditions. One scenario was set for the conditions in North China Plain (NCP) in winter, and the other was set for the conditions in the southeastern United States in summer (SE-US). Because of the differences in latitude (north for China vs south for the United States) and season (winter for China vs summer for the United States), the difference in temperature between these two scenarios was very large, i.e., **29 K** (269 K for China vs 298 K for the United States) (Table S1 in Zheng's paper). Their purpose was to identify factors leading to the large pH difference between these two specific scenarios (~4.7 units of difference).

Our study compared the annual average pH levels using multiple sites in both countries. In particular, the US sites covers the contiguous United States (not just one site in the southeastern United States). The difference in average temperature between China (NCP) and the contiguous United States is only **2.6 K** (287.4 K in China vs 284.8 K in the United States) (Table S3), which is one order of magnitude lower than the temperature difference in Zheng's study. It is expected that with nearly 30 K difference in temperature in Zheng's settings, the contribution of temperature to pH difference could be much larger than our evaluation.

Our nationwide simulations showed that the temperature difference between the US and China is about 5 K in terms of spatial averages (US minus China) and -1.4 K in terms of population-weighted averages (Table S4). In general, the temperature difference of 29 K between Zheng's scenarios is not representative of the temperature difference between China and the United States on an annual average level.

In response to the reviewer's comment, we added temperature and relative humidity as two additional driving factors of pH into our Multivariable Taylor Series Method (MTSM) analysis. Results were updated correspondingly (see details below). We also evaluated the pH difference between the scenarios adopted in Zheng's study (i.e., NCP and SE-US). The results showed that the temperature accounted for 1.3 units of difference in aerosol pH between their two scenarios (see Figure S9 below), which was in line with what was reported in Zheng's paper (1.6 units).

The sentence in line 174 was revised as follows to clarify the inclusion of meteorological variables,
"*To separate the contributions of individual components (eight species in total, including $Na^+$, $SO_4$, $TNO_3$, $TNH_3$, $TCl$, $Ca^{2+}$, $K^+$, and $Mg^{2+}$) and meteorological variables (RH and temperature) to the pH difference ...*"

Results and discussion associated with temperature were added to lines 335-340 as follows,
"*Studies have identified an important role of temperature in driving aerosol pH (Battaglia et al., 2017; Tao and Murphy, 2019; Jia et al., 2020). Our MTSM analysis showed that temperature accounted for 0.07–0.39 unit of pH difference between China and the US, which varies by group (Fig. 6). Such relatively small contributions of temperature, compared to those of $TNH_3$ and $SO_4$, are mainly because of the small difference in temperature between these two countries which are at similar latitudes. The difference in the annual average temperature between China and the US is 1.4 K, -5.0 K, and 2.6 K in the observation, non-weighted, and population-weighted groups, respectively (Table S4).*"

The comparison between results of the contribution of temperature is also discussed in a new section (Sect. 3.2.3 in the revised manuscript, two pathways leading to the aerosol acidity difference) in lines 425-435 as follows,

"*Our results, showing the importance of both mass concentration associated with LWC and chemical composition associated with $H_{air}^+$ and a minor role of temperature, seem in some aspects to contradict a previous study (Zheng et al., 2020) which highlighted LWC and temperature instead of chemical composition as the most important factors explaining the pH difference between China (NCP) and the US. We note that the difference in the conclusions is reasonable when considering the differences in the specific cases examined in these two studies. The previous study compared the conditions in NCP in winter with those in the southeastern US in summer (SE-US). Because of the differences in latitude (north for China vs south for the US) and season (winter for China vs summer for the US), the difference in temperature between their scenarios (29 K) was an order of magnitude greater than those in our study which has greater spatial and temporal coverage (2.6 K in the observation group, 5 K in the non-weighted group, and -1.4 K in the population-weighted group). Using MTSM, we evaluate the pH difference between NCP and SE-US scenarios considered in the previous study. The results show that temperature accounts for 1.3 units of difference in aerosol pH between their two scenarios (Fig. S9), in line with what was previously reported (1.6 units).*"

Figure 6 was updated as follows to include meteorological factors in response to this comment,

[Figure]

**Figure 6.** Contributions of individual components and meteorological factors to (a) total difference of aerosol pH ($\Delta pH$), (b) the aerosol pH difference through the pathway of LWC ($\Delta pH_{LWC}$), (c) the aerosol pH difference through the pathway of H$^+_{air}$ ($\Delta pH_{H_{air}^+}$) between China and the United States calculated by Multivariable Taylor Series Method (MTSM) as described in Sect. 2.4. For each factor, the sum of the contributions through the two pathways yields the net contribution of this factor to the aerosol pH difference. The case in the United States is chosen as the starting point, and China as the ending point.

Figure S9 was added to **Supplementary Information** as follows to provide the results from the sensitivity test using Zheng's settings,

[Figure]

**Fig. S9** Contributions of individual components and meteorological factors to (a) total difference of aerosol pH ($\Delta pH$), (b) through the pathway of LWC ($\Delta pH_{LWC}$), (c) through the pathway of $H^+_{air}$ ($\Delta pH_{H^+_{air}}$) calculated by Multivariable Taylor Series Method (MTSM) between the NCP scenario and the US-SE scenario in Zheng's study (Zheng et al., 2020). For individual factors, the sum of the contributions through the two pathways yields the net contribution of this factor to aerosol pH. The case in the United States is chosen as the starting point, and China as the ending point.

**Aerosol liquid water content (LWC).** The reviewer's comment is well taken. As pointed out by the reviewer, the present manuscript does account for LWC differences because concentration, composition, and meteorology all affect LWC. In the revised paper, we added more analyses and discussion regarding LWC in response to the reviewer's comment. Specifically, we used MTSM to identify factors contributing to LWC difference as what we have done for pH. Our simulations showed a LWC difference of **8.2 μg·m⁻³** between China (NCP) and the contiguous United States in group "observation" in our study and **340 μg·m⁻³** between Zheng's scenarios (NCP minus US-SE). In both simulations, MTSM identified $SO_4^{2-}$, $TNO_3$, and $TCl$ as three most important components leading to the LWC difference (Fig. 6, Fig. S7, Fig. S9). Relative humidity (RH) in our cases played an important role (Fig. 6). RH levels were identical between Zheng's scenarios and thus didn't contributed to any LWC difference (Figure S9).

The much larger LWC difference between Zheng's scenarios than that between ours was mainly driven by the differences in pollutants concentrations. For example, the $SO_4$ concentration is as high as 156 μg·m⁻³ in Zheng's NCP scenario, while only 9.2 μg·m⁻³ in ours. Such differences in concentrations are reasonable, given that Zheng's paper selected a severe haze event occurring in Beijing in winter 2013 as the scenario for China (NCP), while we used annual average levels over NCP in 2017 as our case for China (NCP). Note that winter 2013 was a period when air pollution reportedly reached record high levels across the northern China. Since 2013, China has launched strict controls on air pollutant emissions, and $PM_{2.5}$ levels decreased significantly between 2013 and 2017 (Zhang et al., 2019). Therefore, Zheng's NCP scenario should be more representative of short-term haze events in pre-2013 period, while our China (NCP) case should be more representative of annual average levels in recent years.

Inspired by the reviewer, we noted that there are two pathways through which factors affect aerosol pH—by modifying aerosol LWC and by modifying $H^+$. As pH is calculated as [$\log_{10}$(LWC) − $\log_{10}$($H_{air}^+$) − 3] (LWC and $H_{air}^+$ are expressed as mass per unit volume of air, μg m⁻³), no matter how complex the aerosol system is, the linkage between a perturbation in input factors and the change in pH is either through the change in LWC or in $H_{air}^+$ or both. Based on this view, we updated Figure 6 to include these two pathways associated with LWC and $H^+$, respectively, (see Figure 6b and c as given above). Related results and discussion were provided in Section 3.2.3 as follows,

[revised manuscript text omitted]

Figs. S7 and S8 mentioned in this section are provided in Supplementary information as follows,

[Figure]

**Fig. S7 Step-specific contributions of individual factors to the pH difference between China and the US.** (a), (b), and (c) show the contributions of individual components and meteorological factors to (a) total difference of aerosol pH ($\Delta pH$), (b) through the pathway of LWC ($\Delta pH_{LWC}$), (c) through the pathway of H$^+_{air}$ ($\Delta pH_{H^+_{air}}$) between the US case and an intervening case with the concentrations of all components in the US case multiplied by a constant factor of 8.4. The former is chosen as the starting point, and the latter is chosen as the ending point. (d), (e), and (f) show the contributions of individual components and meteorological factors to (d) total difference of aerosol pH ($\Delta pH$), (e) through the pathway of LWC ($\Delta pH_{LWC}$), (f) through the pathway of H$^+_{air}$ ($\Delta pH_{H^+_{air}}$) between the intervening case and the China case. The former is chosen as the starting point, and the latter is chosen as the ending point. The inputs are shown in Table. S4, sensitivity test.

[Figure]

**Fig. S8** Sensitivity tests showing the pH changes in response to different levels of SO$_4$ and TNO$_3$ in an NH$_4^+$–SO$_4^{2-}$–NO$_3^-$–H$_2$O aerosol system. (a) The pH of an aerosol with fixed TNH$_3$ (2 μg m$^{-3}$) and varied SO$_4$ and TNO$_3$ (from 0.5 μg m$^{-3}$ to 5 μg m$^{-3}$) (b) The pH after multiplying all the inputs in (a) by a factor of 8.4. Note that the SO$_4$ and TNO$_3$ levels shown along the axes are the initial levels before multiplication. (c) pH differences between (b) and (a) (b minus a).

**Nonvolatile cations (NVCs).** Thank you for pointing this out. We found a mistake in our previous calculation where Ca$^{2+}$ was wrongly treated. With the Ca$^{2+}$ treatment being corrected, the

contributions of NVCs were on average one quarter of the previous contributions and therefore, became minor. The results on NVCs were updated throughout the manuscript, as specified below.

The sentences in lines 325-327 (lines 307-313 in the original version) was revised as "*Other cations, mainly NVCs, have a relatively small effect (0.2, 0.2, and 0.3 in group "observation", "simulation", "simulation-weighted", respectively), which is consistent with a previous study (Zheng et al., 2020).*"

The sentence in line 329 (line 316 in the original version), "…was fully offset by $TNH_3$ and NVCs" was revised as "*…is fully offset by $TNH_3$.*"

The sentence in line 428 in the original version, "The MTSM method further shows a significant contribution of NVCs on the pH difference," was removed.

We thank the reviewer again for providing this constructive comment and insightful suggestions.
* * *
**Comment 4**

Finally, there is quite a bit of space (both figures and discussion) dedicated to analyses that don't seem to add much to the manuscript. For example, Line 350 describes the process for segregating the data into different groups to further examine the effects of TNH3. This was a good idea, however, the results (shown in Fig. 8) don't add any new insight to our current understand of aerosol pH. I would say the same is true for Fig. 11 and the associated discussion, and for the analysis of the TNO3/TSO4 molar ratios. I would strongly suggest moving these figures and discussion to the Supporting Information, especially in light of the added discussion and possible analyses needed to address the above comments.

**Response**

We thank the reviewer for providing this suggestion. In response to this comment, **Section 3.2.3 Effects of ammonium on aerosol pH**, **3.2.4 The relationship between sulfate/nitrate and aerosol pH**, and Figures 7-11 in the original manuscript were moved to **Supplementary Information (Text 1, Text 2 and Figs. S10-S14)**. The following sentence was added to lines 333-334 to provide a link.

"*More detailed analyses and discussions on the effects of $TNH_3$, $TNO_3$, and $SO_4$ on aerosol pH can be found in Supplementary Information.*"

**Abstract** and discussion in the main text were updated correspondingly.
* * *
**Comment 5**

Line 51: delete the period appearing in the middle of the sentence.

**Response:**

Deleted.
* * *
**Comment 6**

Line 62-63: what are large-scale measurements?

**Response**

To avoid confusion, "large-scale" was removed in the revised version.
* * *
**Comment 7**

Line 93. I question the stated accuracy of the AMoN NH3 measurements – especially given the variability between duplicate samples reported by the network.

**Response:**

Thanks so much for pointing this out. We are sorry for making this mistake. In response to this comment, the sentence was revised as follows (lines 91-95 in the revised manuscript),

"*The accuracy of CASTNET measurements has been assessed through the analysis of reference and continuing calibration verification samples with a criterion of 95-105% (except $NH_4^+$, whose accuracy criterion is 90-110%) (United States Environmental Protection Agency, 2012). Detailed information about data quality is available in the CASTNET Quality Assurance Report-Annual 2011 (United States Environmental Protection Agency, 2012). A previous study demonstrated that the $NH_3$ concentrations measured by the passive AMoN samplers are comparable to annular denuder systems (as a reference system) with a mean relative percent difference of -9% (Puchalski et al., 2015).*"
* * *
**Comment 8**

Line 97: "Its" should not be capitalized

**Response:**

Corrected in the updated manuscript. Thank you.
* * *
**Comment 9**

Provide the criteria for identifying outliers, and the number of outliers excluded from the respective datasets.

**Response**

Thank you for the suggestion. In this study, we selected data very carefully. Provided continuous measurements, there should be 5840 cases in China because we have daily data in one-year period at 16 monitoring sites. However, data in many cases are missing due to the interruption during measurements (e.g., there are no data in January at monitoring sites 1, 3, 4, 5). Therefore, the first step was to remove the cases with missing data of any of the measured species. In this step, 3136 cases were left. Then we identified the outliers as the data beyond the scope of three times the median absolute deviations (MAD) from the median for each component. We removed the cases with any component identified as an outlier. Eventually, we got 1766 cases with valid data of all components. Although we removed a large number of cases in this process, the remaining data covered most of the time in a year and generally distributed evenly by months (see Table S3 below).

In response to this comment, we added description of this data selection process in the main text in lines 100-108 as follows,

"*We derive daily average concentrations of gaseous species including $NH_3$, $HNO_3$ and HCl and of particle species including $NH_4^+$, $NO_3^-$, $Cl^-$, and NVCs from hourly observational data at 16 monitoring sites for use in pH calculation. These monitoring sites are clustered in NCP in eastern China (Fig. S2c). Due to the lack of data quality information, we first process the data by removing unreasonable data points. We define a set of valid data containing all the measured components in one day as one case. We first remove cases with one or more missing components. In this step, 2704 of 5840 cases are removed. We then identify data points that are more than three median absolute deviations from the median as outliers and remove cases with any component identified as an outlier. Eventually, 1766 cases remain for subsequent analyses. Although we remove many cases in this process, the remaining cases cover most of the days in a year and are evenly distributed by month (Table S3).*"

Table S3 was added to Supplementary Information as follows,

**Table S3 Distribution of observational cases in China in each month (with outliers removed)**

| Month | Jan. | Feb. | Mar. | Apr. | May | Jun. | Jul. | Aug. | Sep. | Oct. | Nov. | Dec. |
|---|---|---|---|---|---|---|---|---|---|---|---|---|
| Number of cases | 104 | 144 | 176 | 157 | 268 | 184 | 182 | 111 | 30 | 45 | 134 | 231 |

**Comment 10**

Line 158-159: specify if sulfate was also adjusted.

**Response**

In response to this comment, the sentence in line 162 (line 158-159 in the original version) was revised as follows to clarify that sulfate was not adjusted.

*"In order to avoid this potential bias, we use modified $Ca^{2+}$ concentration for pH calculations while keeping $SO_4^{2-}$ concentration unchanged…"*

**Comment 11**

Line 241-241: need to acknowledge that most of the pH predictions over China cannot be evaluated due to limitations in observational data.

**Response**

In response to this comment, the following sentence was added in line 260 to clarify this point,
"*It should be noted that due to the scarcity of observational data, the pH estimates in southern and western China are not evaluated.*"

**Comment 12**

Line 251-252: these correlations are weak, so the description of a "significant positive correlation" is misleading.

**Response**

Statistically, to determine whether the correlation between variables is "significant", we compare the *p* value with a significant level (*a*) (https://support.minitab.com/en-us/minitab-express/1/help-and-how-to/modeling-statistics/regression/how-to/correlation/interpret-the-results/). Typically, *a* can be 0.05 or 0.01.

As given in the sentence in lines 269-270 (lines 251-252 in the original version), the *p* values are less than 0.0001 (lower than either 0.05 or 0.01). Therefore, the correlations mentioned here are significant. In the revised sentence, we added the word "*statistically*" before "significant" and provided *a* as follows,

"*This finding is further confirmed by the statistically significant positive correlation, …(China: r=0.42, p<0.0001; the United States: r=0.28, p<0.0001) (a=0.01).*"

**Comment 13**

Line 325: "this could be due to higher biases in H+ concentration by ISORROPIA in ammonia poor conditions" – I don't follow this explanation?

**Response**

Sorry for the confusion. After checking the input of that figure, we found that we did not used the most updated input where we modified the $Ca^{2+}$ concentration as we mentioned in Sect. 2.3, line 164. After the modification, the pH change is more stable, so we deleted the original statement. Fig. 7 (Fig. S10 in the updated manuscript) was updated as follows,

[Figure]

**Fig. S10 Responses of pH, ε(NH₄⁺) and ε(NO₃⁻) to the change of TNH₃ from 0.1 to 1000 μg·m⁻³ while keeping all other components constant at their annual average levels.** The shaded areas show the TNH₃ concentration ranges that covers 75% of the observed cases, the dashed lines show the 5th and 95th percentiles of the observed cases. The squares and diamonds mark the average TNH₃ levels in China and the United States, respectively.
* * *
**Comments 14**

Line 457: awkward as written

**Response**

This sentence was removed from the revised manuscript. The language of the manuscript has been checked by two native English speakers.

Again, we thank the reviewer for their overall support and thoughtful suggestions. These suggestions have helped improve the quality of the paper substantially.

**References**

[revised manuscript text omitted]

---

## Referee Report (RR1)

Review of the manuscript revision, "Significant contrasts in aerosol acidity between China and the United States"

Overall, the authors have done an excellent job addressing referee comments. The updated manuscript is greatly improved and is nearly ready for publication. I do have one substantive comment and several minor technical corrections. All of the comments are straightforward to address.

In the revised analysis, one of the findings, in particular, stood out to me as quite novel and significant. This analysis (discussed in lines 385 – 405 and presented in Figure S7) found that pH changed when the concentrations of all chemical components were scaled by a common factor, keeping meteorological parameters constant. I agree with the authors that this is a surprising result. Although the authors cite several references in lines 428-429 that discuss the effects of concentration and chemical composition on pH, none of those studies showed this specific result (that pH changed with mass concentration even though all chemical component mole fractions did not change). I think that the authors should consider highlighting this result more prominently in the paper, e.g., in the abstract.

Technical Corrections:
- Throughout the manuscript, the 'dot' in $\mu g \ m^{-3}$ should be removed.
- In Figures 1, 4, and 5, change figure labels from "the United States" to "United States"
- Line 302: delete "process"
- Line 389: comma after "pH"
- Sentence beginning on line 390 should be edited for grammar.
- Entire paragraph beginning on line 395 should be edited for grammar.
- Line 407: what is meant by "weak acidic capacity"? suggest revising this sentence.
- Line 411: change "sensitive" to "sensitivity"
- Line 417: delete "found"
- Line 418-419: awkward as written – suggest revising this sentence
- Line 427: what is meant by "availability of the corresponding aerosol components"? suggest revising this sentence
- Line 440: comma after "study"
- Line 460: comma after "SO4"
- Line 462: "linking to the pH difference" is awkward - suggest revising this sentence

---

## Author Response (AR2)

**Response to Anonymous Referee #2**
Manuscript: *Significant contrasts in aerosol acidity between China and the United States*
Manuscript number: acp-2020-879
Journal: Atmospheric Chemistry and Physics
Authors: Bingqing Zhang, Huizhong Shen, Pengfei Liu, Hongyu Guo, Yongtao Hu, Yilin Chen, Shaodong Xie, Ziyan Xi, T. Nash Skipper, Armistead G. Russell
* * *
**Comment 1**

Overall, the authors have done an excellent job addressing referee comments. The updated manuscript is greatly improved and is nearly ready for publication. I do have one substantive comment and several minor technical corrections. All of the comments are straightforward to address.

**Response**
We would like to thank the reviewer for the time reviewing our manuscript and providing supportive comments. We have listed the point-by-point responses below. We hope that our revision and this new version of the manuscript have addressed all the reviewer's concerns.
* * *
**Comment 2**

In the revised analysis, one of the findings, in particular, stood out to me as quite novel and significant. This analysis (discussed in lines 385 – 405 and presented in Figure S7) found that pH changed when the concentrations of all chemical components were scaled by a common factor, keeping meteorological parameters constant. I agree with the authors that this is a surprising result. Although the authors cite several references in lines 428-429 that discuss the effects of concentration and chemical composition on pH, none of those studies showed this specific result (that pH changed with mass concentration even though all chemical component mole fractions did not change). I think that the authors should consider highlighting this result more prominently in the paper, e.g., in the abstract.

**Response**
In response to this comment, we highlighted the results of pH change with mass concentrations by rewriting the following sentences in abstract in lines 24-29.

*"Our assessment shows that the differences in mass concentrations and chemical composition play equally important roles in driving the aerosol pH difference between China and the US — increasing the aerosol mass concentrations, but keeping the relative component contributions the same, in the US to the level in China (by a factor of 8.4) increases the aerosol pH by ~1.0 unit, and further shifting the chemical composition from US conditions to China's that is richer in ammonia increases the aerosol pH by ~0.9 units."*
* * *
**Technical Correction 1**

- Throughout the manuscript, the 'dot' in µg m-3 should be removed.

**Response**
Removed throughout.
* * *
**Technical Correction 2**

- In Figures 1, 4, and 5, change figure labels from "the United States" to "United States".

**Response**
Changed in response to this comment.
* * *
**Technical Correction 3**

- Line 302: delete "process".

**Response**
Deleted in response to this comment.
* * *
**Technical Correction 4**

- Line 389: comma after "pH"

**Response**
Added in response to this comment.
* * *
**Technical Correction 5**

- Sentence beginning on line 390 should be edited for grammar.

**Response**
In response to this comment, the sentence was revised as follows,
"*The second step that changes the chemical composition shows a further increase of 0.76 units in the aerosol pH, which is mainly achieved through the $H_{air}^+$-modifying pathway (0.88 units). The LWC-modifying pathway plays a minor role (-0.11 unit) in this step (Fig. S7 (d), (e), (f)).*"
* * *
**Technical Correction 6**

- Entire paragraph beginning on line 395 should be edited for grammar.

**Response**
In response to this comment, the paragraph was edited for grammar and revised as follows,
"*It is surprising, in the first step, that pH changed when the concentrations of all chemical components were scaled by a common factor. This means that pH changes with mass concentration of the aerosol (gas+particle) even though all chemical component mole fractions hold. Further investigation shows that, increasing the aerosol mass concentration drives $TNO_3$ and $TNH_3$ partitioning toward particle phases—$\varepsilon(NH_4^+)$ and $\varepsilon(NO_3^-)$ increase from 0.4 to 0.6 and from 0.6 to 0.98, respectively. Given the weak acidity of $NO_3^-$, the particle is ultimately neutralized by the increased $NH_4^+$. The repartitioning in response to the increase in mass concentration is thus key to the pH shift and can be explained by Henry's Law, i.e., $[A_{aq}]=H_A \cdot p_A$, where $[A_{aq}]$ is the aqueous-phase concentration of component A in the unit of moles per liter water, $p_A$ is the partial pressure of A in the gas phase, and $H_A$ is Henry's law coefficient (Seinfeld and Pandis, 2006). $[A_{aq}]$ is proportional to $c_A$ / LWC ($c_A$ denotes the particle-phase concentration of A, note that LWC and $c_A$ are expressed as mass per unit volume of air, and $[A_{aq}]$ is expressed as moles per unit volume of water). Increasing the concentrations of all chemical components by a*

*common factor increases $p_A$ (due to the increase in the gas-phase concentration of A) but does not change [$A_{aq}$] (because both $c_A$ and LWC increases in the same direction by the same magnitude). According to the Henry's Law, more gas-phase A will thus shift toward the particle phase to achieve a thermodynamic equilibrium of the new system.*"

**Technical Correction 7**

- Line 407: what is meant by "weak acidic capacity"? suggest revising this sentence.

**Response**

We have replaced "weak acidic capacity" with "weak acidity" in the revised manuscript.

**Technical Correction 8**

- Line 411: change "sensitive" to "sensitivity".

**Response**

Changed in response to this comment.

**Technical Correction 9**

- Line 417: delete "found".

**Response**

Deleted in response to this comment.

**Technical Correction 10**

- Line 418-419: awkward as written – suggest revising this sentence

**Response**

In response to this comment, this sentence was revised as follows,

"*In populated continental regions, mass fractions of TNH₃ are often high (Bencs et al., 2008; Behera and Sharma, 2010; Zheng et al.,2015; Cheng et al., 2016; Guo et al., 2017b), and an increase in mass concentration thus typically increases the aerosol pH.*"

**Technical Correction 11**

- Line 427: what is meant by "availability of the corresponding aerosol components"? suggest revising this sentence.

**Response**

In response to this comment, "availability" was changed to "fractions".

**Technical Correction 12**

- Line 440: comma after "study".

**Response**

Added in response to this comment.

**Technical Correction 13**

- Line 460: comma after "SO4".

**Response**

Added in response to this comment.

**Technical Correction 14**

- Line 462: "linking to the pH difference" is awkward - suggest revising this sentence.

**Response**

In response to this comment, this sentence was revised as "*Further investigation highlights two pathways related to the pH difference—one associated with changes in LWC and the other associated with changes in $H_{air}^+$.*"